# Profound regulation of Na/K pump activity by transient elevations of cytoplasmic calcium in murine cardiac myocytes

Fang-Min Lu, Christine Deisl, Donald W Hilgemann*

Department of Physiology, University of Texas Southwestern Medical Center at Dallas, Dallas, United States

**Abstract** Small changes of Na/K pump activity regulate internal Ca release in cardiac myocytes via Na/Ca exchange. We now show conversely that transient elevations of cytoplasmic Ca strongly regulate cardiac Na/K pumps. When cytoplasmic Na is submaximal, Na/K pump currents decay rapidly during extracellular K application and multiple results suggest that an inactivation mechanism is involved. Brief activation of Ca influx by reverse Na/Ca exchange enhances pump currents and attenuates current decay, while repeated Ca elevations suppress pump currents. Pump current enhancement reverses over 3 min, and results are similar in myocytes lacking the regulatory protein, phospholemman. Classical signaling mechanisms, including Ca-activated protein kinases and reactive oxygen, are evidently not involved. Electrogenic signals mediated by intramembrane movement of hydrophobic ions, such as hexyltriphenylphosphonium (C6TPP), increase and decrease in parallel with pump currents. Thus, transient Ca elevation and Na/K pump inactivation cause opposing sarcolemma changes that may affect diverse membrane processes.

*For correspondence: donald. hilgemann@utsouthwestern.edu

Competing interests: The authors declare that no competing interests exist.

## Introduction

Na/K pumps establish the Na and K gradients used for ion transport and electrical signaling in animal cells (*Skou, 1990*), thereby accounting for a large fraction of total energy expenditure (*Milligan and McBride, 1985*). In cardiac myocytes, small changes of pump activity have large effects on excitation-contraction coupling by shifting the energetic equilibrium of Na/Ca exchangers (*Reuter et al., 2002*; *Hilgemann, 2004*). Accordingly, regulatory mechanisms that control Na/K pump activity are of high biological significance.

Surprisingly, cardiac Na/K pump regulation remains rather enigmatic. The FXYD protein, phospholemman, can inhibit Na/K pumps with analogy to phospholamban action at SERCA Ca pumps, and this inhibition is proposed to be released when phospholemman is phosphorylated by cAMP-dependent protein kinases (PKAs) (*Bibert et al., 2008*; *Han et al., 2010*; *Mishra et al., 2015*). Interestingly, phosphorylation of phospholemman by PKAs appears to increase the affinity of pumps for cytoplasmic Na, while phosphorylation of phospholemman by protein kinase Cs (PKCs) increases maximal pump activity (*Han et al., 2006*). These effects appear robust in optical experiments with Na-sensitive dyes, but without control of membrane potential, while the equivalent results in patch clamp-controlled experiments are remarkably inconsistent. During whole cell patch clamp of cardiac myocytes, activation of PKAs has been reported to duplicate results with dyes (*Despa et al., 2005*), to mildly stimulate pump currents under all experimental conditions (*Kockskamper et al., 2000*), to have no effect (*Ishizuka and Berlin, 1993*; *Main et al., 1997*; *Fine et al., 2013*), or to inhibit pump currents via an oxidative mechanism (*Galougahi et al., 2013*). One analysis suggests that

**eLife digest** All animal cells have pumps in their outer membrane that continuously pump out sodium ions and bring in potassium ions. As a result, the concentration of sodium ions inside cells is low in comparison to the concentration outside, and vice versa for potassium ions. These differences between inside and outside concentrations are a source of energy for cells to do a variety of important tasks, similar to using energy that is stored in a reservoir formed by a dam. When cells allow ions to move in the direction they naturally move, similar to water moving over a dam, they carry out two important roles. First, they generate electrical signals that are the basis of all fast communication in nerves and muscles. Second, they can force important molecules to be moved into or out of cells in a specific way, thereby making the inside composition of cells different from the outside.

Lu et al set out to uncover how the pumps that move sodium ions out of and potassium ions into animal cells are regulated. The experiments focused on heart muscle cells from mice, and revealed that sodium-potassium pumps regulate themselves via a self-inhibitory mechanism that depends on the sodium concentration in cells. This "auto-inhibition" reaction appears to affect the activity of neighboring membrane proteins that transport sodium ions. Further experiments showed that the auto-inhibition reaction is controlled by calcium-dependent processes that change the physical properties of the cell's membrane.

The next challenges will be to determine how sodium transporters influence one another and how the calcium signals actually alter the physical properties of the surface membrane.

phosphorylation by PKA *inhibits* pump activity in the absence of Ca, but overcomes an inhibitory action of Ca and thereby *enhances* pump activity, albeit modestly, in the presence of cytoplasmic Ca (*Gao et al., 1996*). Results for PKCs are similarly complex and presumably reflect complexities of PKC signaling in cardiac myocytes. Both small stimulatory effects (*Gao et al., 1999*; *Han et al., 2006*) and inhibitory effects (*White et al., 2009*) are reported, the latter occurring through oxidative stress signaling mechanisms. Na/K pumps are evidently regulated by trafficking mechanisms in some cell types (*Al-Khalili et al., 2003*; *Liu and Shapiro, 2007*; *Lecuona et al., 2009*; *Alves et al., 2015*), but there is little support for physiological regulation of cardiac pumps by these mechanisms. Non-conventional endocytic mechanisms can remove pumps from the sarcolemma during reperfusion injury (*Lin et al., 2013*). Other mechanisms proposed to regulate the activity of Na/K pumps in cardiac myocytes include redox-dependent glutathionylation of the beta subunits of Na/K pumps (*Liu et al., 2012*) and the modulation of phospholemman function by palmitoylation (*Tulloch et al., 2011*).

Given this background, we initiated a new analysis of pump regulation in murine myocytes, starting with the observation that Na/K pump currents can run down during whole-cell patch clamp experiments. Attempting to stop and/or reverse this run-down, it became evident that a brief elevation of cytoplasmic Ca via reverse Na/Ca exchange had substantially larger effects on pump activity than the activation of either PKAs or PKCs. As described here, the stimulatory effects of transient Ca elevation, usually associated with spontaneous beating, occur as an apparent increase of the cytoplasmic Na affinity of Na/K pumps, the same functional effect proposed to occur with PKA activation (*Despa et al., 2005*). Subsequent to a transient Ca elevation, peak pump currents activated by extracellular K are strongly enhanced, and the current decay that occurs during continued K application is attenuated. This decay has been ascribed previously to depletion of cytoplasmic Na in a restricted space (Fujioka et al.; *Su et al., 1998*; *Verdonck et al., 2004*), and we explore here the possibility that current decay reflects instead, or additionally, an inactivation mechanism that bears similarity to the Na-dependent inactivation of Na/Ca exchangers (*Hilgemann et al., 1992b*). In contrast to Na/Ca exchangers, inactivation of Na/K pumps would not occur when cytoplasmic binding sites are occupied by Na. Rather, it would occur when one or more Na binding sites are unoccupied, while recovery from inactivation would be promoted by Na binding. We do not discern any effect of PKA activation in the absence of Ca elevations and the stimulatory effects of Ca elevation occur similarly in phospholemman-deficient myocytes. Further results suggest that physical changes of the

membrane itself may be involved, rather than classical signaling mechanisms, and that Na transporters in close vicinity to one another may interact functionally by modifying physical properties of the adjacent bilayer.

## Results

### Overview of Na/K pump function and inactivation hypothesis

To facilitate the presentation of experiments and their interpretation, we present first in *Figure 1 a* cartoon of the Na/K pump cycle as it is widely thought to occur, together with our inactivation hypothesis: The normal cycle consists of a series of reactions in which 3 Na ions are bound from the cytoplasmic side when pumps are in the 'E1' configuration (i.e. with binding sites open to the cytoplasmic side), followed by phosphorylation of the pump, Na occlusion and deocclusion to the outside in the 'E2' configuration, binding of 2 extracellular K ions, occlusion of the K ions, dephosphorylation of the pump, and deocclusion of K to the cytoplasmic side in the E1 configuration with renewed ATP and Na binding. We hypothesize that Na/K pumps in intact murine myocytes can enter into long-lived inactive states when they are in E1 configurations. Specifically, we hypothesize that pumps inactivate preferentially when the Na-selective binding site (*Kanai et al., 2013*; *Vedovato and Gadsby, 2014*) is not occupied by Na, while recovery from inactivation may be favored by Na binding to all sites. Inactivation could in principle be a process that hinders Na binding and/or hinders pump phosphorylation, therefore preventing the completion of a forward pump cycle. Transient cytoplasmic Ca elevations may act to attenuate Na/K pump inactivation via membrane modifications mediated by lipid-modifying enzymes *or* by long-lived conformational changes of high-density membrane-associated proteins.,

### Na/K pump current decay

*Figure 2* introduces the experimental conditions under which Na/K pump currents in murine cardiac myocytes are strongly activated by brief elevations of cytoplasmic Ca. Results in *Figure 2A to D* are at 37°C, while results in *Figure 2E and F* are at 23°C. The extracellular solution contains predominantly N-methyl-d-glucamine (NMG) as monovalent cation, and aspartate (Asp) is the major anion on both sides. Details of solution compositions, which minimize other currents, are given in Experimental Methods. Pump currents are activated and deactivated by moving the myocytes quickly back and forth between solution streams containing 7 mM Na and 7 mM K. By employing equal monovalent cation concentrations, changes of monvalent cation leak currents are minimized. As will be described in Figure 4, we have verified that the concentrations of K which contaminate the solutions routinely employed are adequate to activate substantial pump activity in nominally Na- and K-free solutions (*Rakowski et al., 1989*). Furthermore, it will be demonstrated in Fig. 4 that the 7 mM Na employed in our standard extracellular solution is adequate to effectively block this activity.

*Figure 2A* is a representative current record when pumps are maximally activated by cytoplasmic Na (110 mM) and extracellular K (7 mM). Maximal pump currents are at least 5 pA/pF. The pump currents typically activate within solution switch times (~100 ms) and remain nearly constant during application of extracellular K. In contrast, *Figure 2B* is a representative current record using a cytoplasmic solution with 90 mM K and 25 mM Na. In the presence of cytoplasmic K and with this lower cytoplasmic Na concentration, the pump current activated by extracellular K (7 mM) amounts to 2.4 pA/pF and immediately begins to decline, approaching a steady state over 10 to 15 s. The decaying current component typically amounts to more than one-half of the peak current. Importantly for our subsequent analysis, membrane capacitance typically decreases abruptly by 0.5 to 1% upon application of extracellular K and then declines further, but to a lesser extent, as pump current falls during K exposure. Upon removal of K, the capacitance signal returns toward its baseline with a slow multi-second time course that mimics in our experience the availability of pump currents for activation by renewed application of extracellular K. To test for sensitivity to heart glycosides, the myocyte was moved into two separate solution streams containing ouabain at a concentration that was high enough (0.3 mM) to effectively block rodent Na/K pumps (*Herzig and Mohr, 1984*). The currents were very effectively inhibited by ouabain (n = 5), demonstrating that they indeed reflect Na/K pump activity. Importantly, the transient capacitive signals, as well as pump currents, are ablated by ouabain, indicating that they likely arise from Na/K pumps.

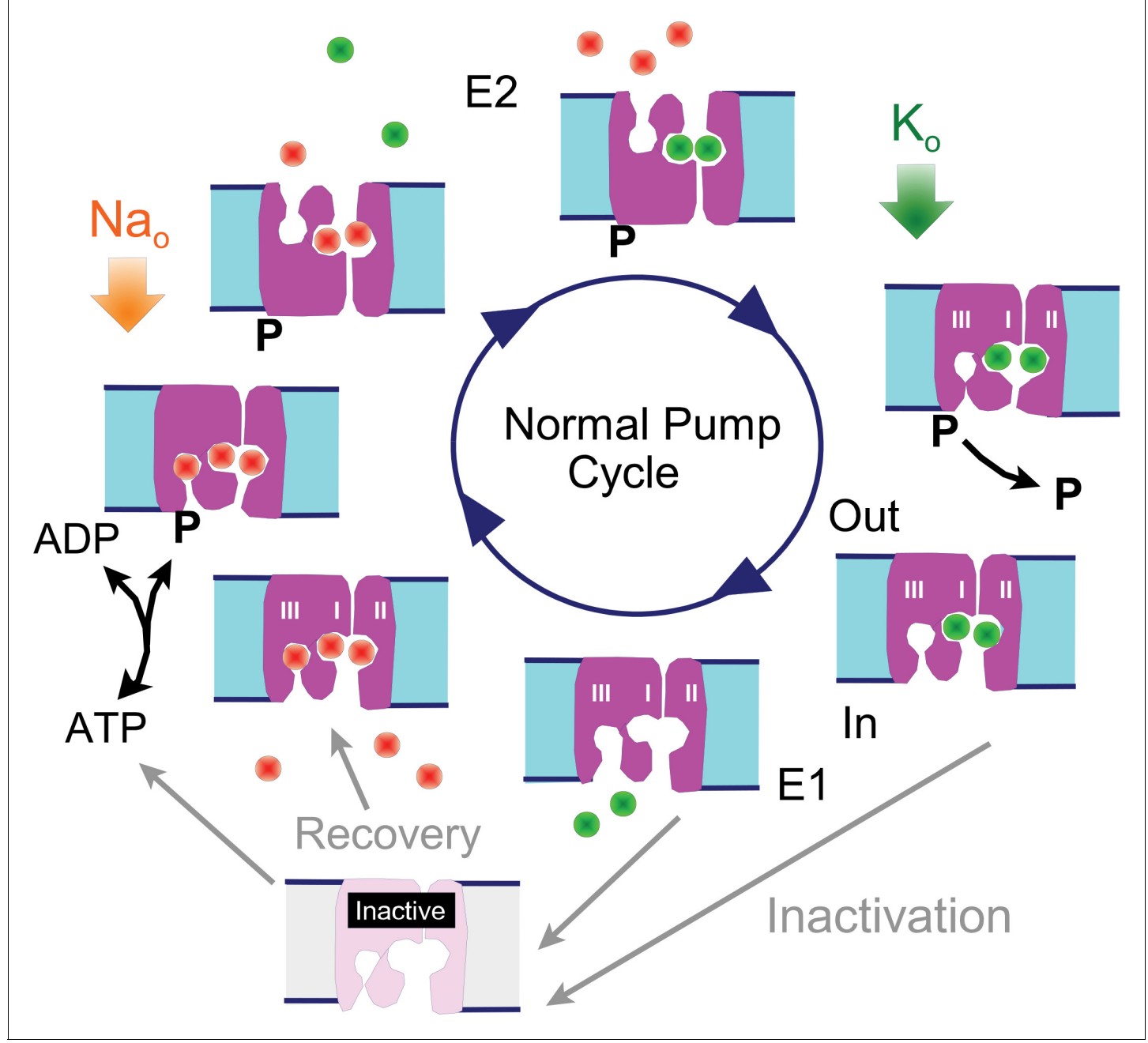

**Figure 1.** Reaction scheme of the Na/K pump with inactivation from E1 states. E1 states are open to the cytoplasmic side and can bind 3 Na ions, Site III being a Na-selective site. Upon phosphorylation the Na ions are occluded and released to the outside in E2 states, followed by binding and occlusion of 2 K ions. Upon dephosphorylation pumps open to the cytoplasmic side in the E1 configuration and release K ions. Hypothetically, Na/K pump current decay occurs as an inactivation reaction taking place preferentially (but probably not exclusively) from 'E1' states in which the Na-selective ion binding site is not occupied by Na. Inactivation might occur with a lower probability from other states, including E2 states. Recovery from inactivation may be promoted by Na binding to inactive exchangers.

As illustrated in the cartoon in *Figure 1*, the absence of extracellular K will cause pumps to accumulate nearly exclusively in the E2 conformation because the E2 to E1 transition is blocked. Application of extracellular K then allows pumps to cycle between E2 and E1 conformations. However, all pumps should return to the E2 configuration immediately upon removing extracellular K. As evident in *Figure 2A*, the capacitive signal returns immediately to the pre-K value when cytoplasmic Na is

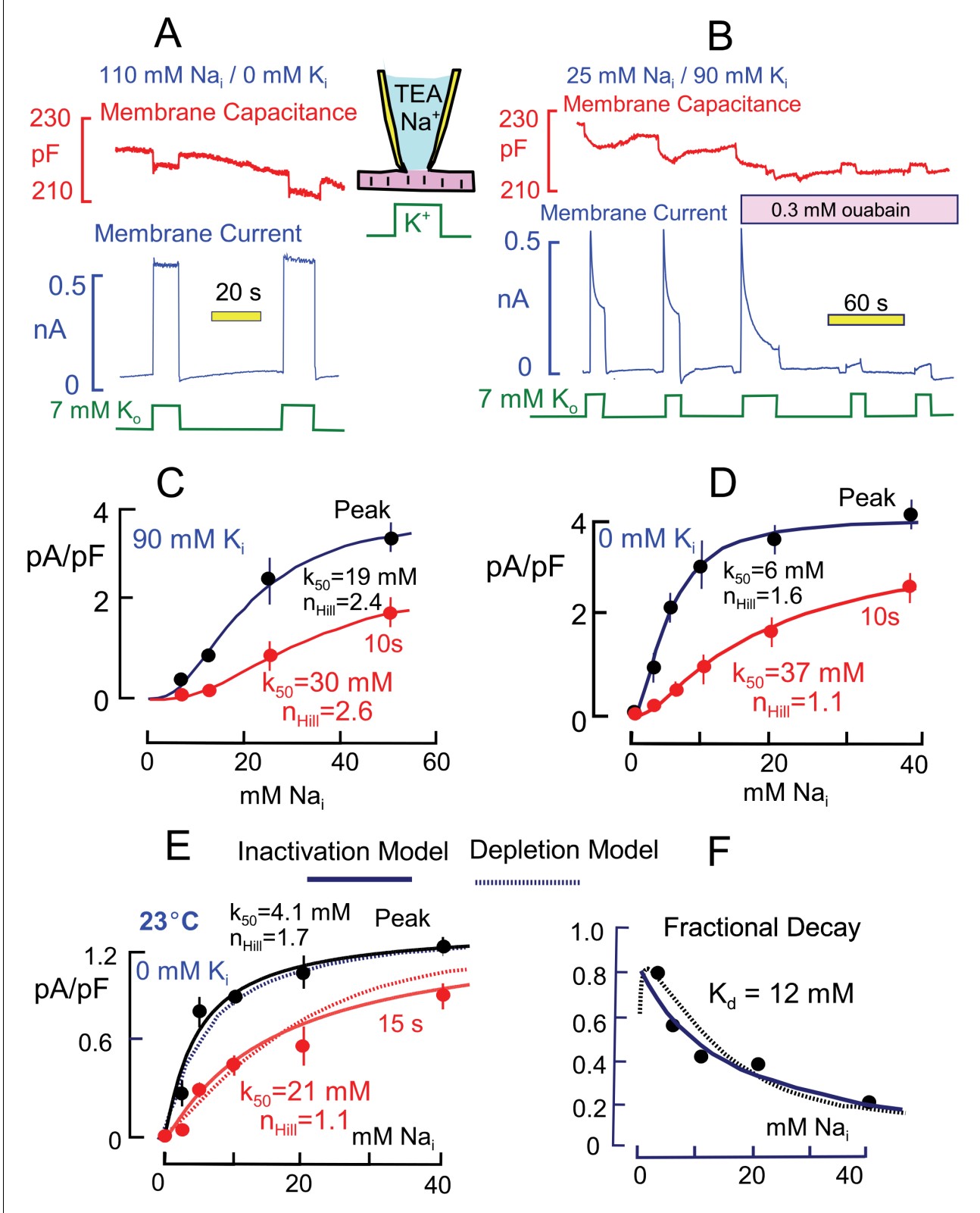

**Figure 2.** Basic properties of Na/K pump currents in murine cardiac myocytes. Solutions contain 20 mM TEA on both sides to minimize K currents, 4 mM Mg on the outside to promote seal formation without Ca, and NMG as the major extracellular cation. Pump currents are initiated by replacing 7 mM Na for 7 mM K on the extracellular side. (A) Outward Na/K pump currents in the presence of 110 mM cytoplasmic Na and no cytoplasmic K are large (>6 pA/pF) and stable. Capacitance decreases and increases by ~1% within solution switch times when K is applied and removed, respectively. (B)

*Figure 2 continued on next page*

*Figure 2 continued*

Outward Na/K pump currents in the presence of 25 mM cytoplasmic Na and 90 mM cytoplasmic K decay over several seconds by more than 50%. Membrane capacitance decreases immediately during K application, as just described, and can then continued to decrease slowly but to a lesser extent during current decay; capacitance recovers slowly over 20 s after removing extracellular K. Typical for rodent Na/K pumps, currents are quite resistant to heart glycosides but can be rapidly and effectively blocked by a high concentration of ouabain (0.3 mM). (**C** and **D**) Cytoplsmic Na dependence of peak and steady state (10 s) pump currents at 37°C *with* and *without* 90 mM cytoplasmic K. (**C**) In the presence of 90 mM K, the $K_{50}$ for peak current is 19 mM Na and the Hill coefficient is 2.4, while the 10 s current shows a $K_{50}$ of 30 mM and Hill slope of 2.6. (**D**) In the absence of cytoplasmic K, peak currents are best described by a Hill equation with a $K_{50}$ of 6 mM cytoplasmic Na and a Hill slope of 1.6. The 10 s current is shifted to a $K_{50}$ of 37 mM with a Hill slope of 1.1 Although currents are smaller in the presence of cytoplasmic K, the fractional decay of current is greater at all Na concentrations. (**E**) Cytoplasmic Na dependence of peak and quasi steady state (15 s) pump currents at 23°C without cytoplasmic K. Data points are the average of values from at least 5 separate myocytes. (**F**) The Na dependence of the fractional decay of pump current (1–15 s/peak) is well described by the occupancy of a single Na binding site with an apparent $K_d$ of 12 mM. The solid and dotted lines in panels **E** and **F** are predictions for overtly simplified Na depletion and inactivation models, described in Materials and methods.

high and pump currents are stable. However, the capacitive signals do not immediately return to baseline after removing extracellular K when cytoplasmic Na is submaximal, as in *Figure 2B*. This a first indication that pumps may become locked into states that cannot cycle during the time in which currents decay. In contrast, the idea that subsarcolemmal Na decreases during pump activity provides no explanation for this pattern.

Concerning the source of these capacitance signals, we have shown previously that in the presence of low extracellular Na concentrations (<20 mM), the rapid binding of Na to E2 configurations of the pump gives rise to significant capacitive signals that dissipate when pumps transition from the E2 to the E1 configuration (*Lu et al., 1995*). We verified that Na binding is a major component of these signals as follows: Measurements were initiated with the standard extracellular solution containing 20 mM Na. Upon switching extracellular Na from 20 mM to lower concentrations and back to 20 mM, capacitance reversibly decreased and increased within solution switch times by about 0.5%, as in *Figure 2A* (5 observations). A significant but smaller involvement of both protons and contaminating K ions remains a possibility. Also, small contributions from still other sources are required to explain the small, gradual declines of capacitance that sometimes occur in the presence of K (e.g. in *Figure 2B*, as compared to *Figure 3B*).

*Figure 2C and D* present the Na dependence of peak and quasi steady state (10 s) Na/K pump currents in the presence and absence of 90 mM K. A somewhat larger Na range was used in experiments with K because the apparent Na affinity is lower than without K. Results were obtained one-myocyte and one-Na-concentration at a time. Peak and 10 s currents were averaged 3 to 4 times in individual myocytes, and the average individual myocyte results were then averaged from at least 4 myocytes under the same conditions for each data point presented. The composite results were fit to Hill functions, and values for the Hill parameters are given in the graphs. The results are similar to an equivalent data set presented by Su et al. (*Su et al., 1998*), and the shift of Na dependence is reminiscent of changes that occur in response to rapid electrical stimulation in skeletal muscle (*Clausen, 2013*). The absence of cytoplasmic K shifts the half-maximal Na concentration of the peak currents from 19 to 6 mM, while the 10 s values shift from 30 to 37 mM. Accordingly, the peak pump current densities are substantially larger at submaximal cytoplasmic Na concentrations in the absence of cytoplasmic K. The Hill coefficient are substantially larger with cytoplamsic K (2.4 and 2.6 for peak and 10 s currents) than without cytoplasmic K (1.6 and 1.1 for peak and 10 s currents).

As noted in the Introduction, multiple studies suggest that the decay of Na/K pump currents during continued application of extracellular K in cardiac myocytes is caused by depletion of submembrane cytoplasmic Na. Assuming that diffusion is relatively temperature-independent, whereas pump activity is strongly temperature-dependent, we analyzed the dependence of peak and quasi steady state (15 s) pump currents at 23°C with the expectation that current decay via Na depletion would be reduced. The experiments were perfomed without cytoplasmic K and are presented in *Figure 2E*. At the lower temperature, pump currents are at least 4-times smaller than at 37°C. Fitting the composite results for peak and 15 s currents to Hill functions, the half maximal pump currents occurred at 4.1 and 21 mM Na, respectively, and Hill slopes were 1.7 and 1.1, respectively, for peak and 15 s currents. The overall pattern of current decay is not impressively different from results at

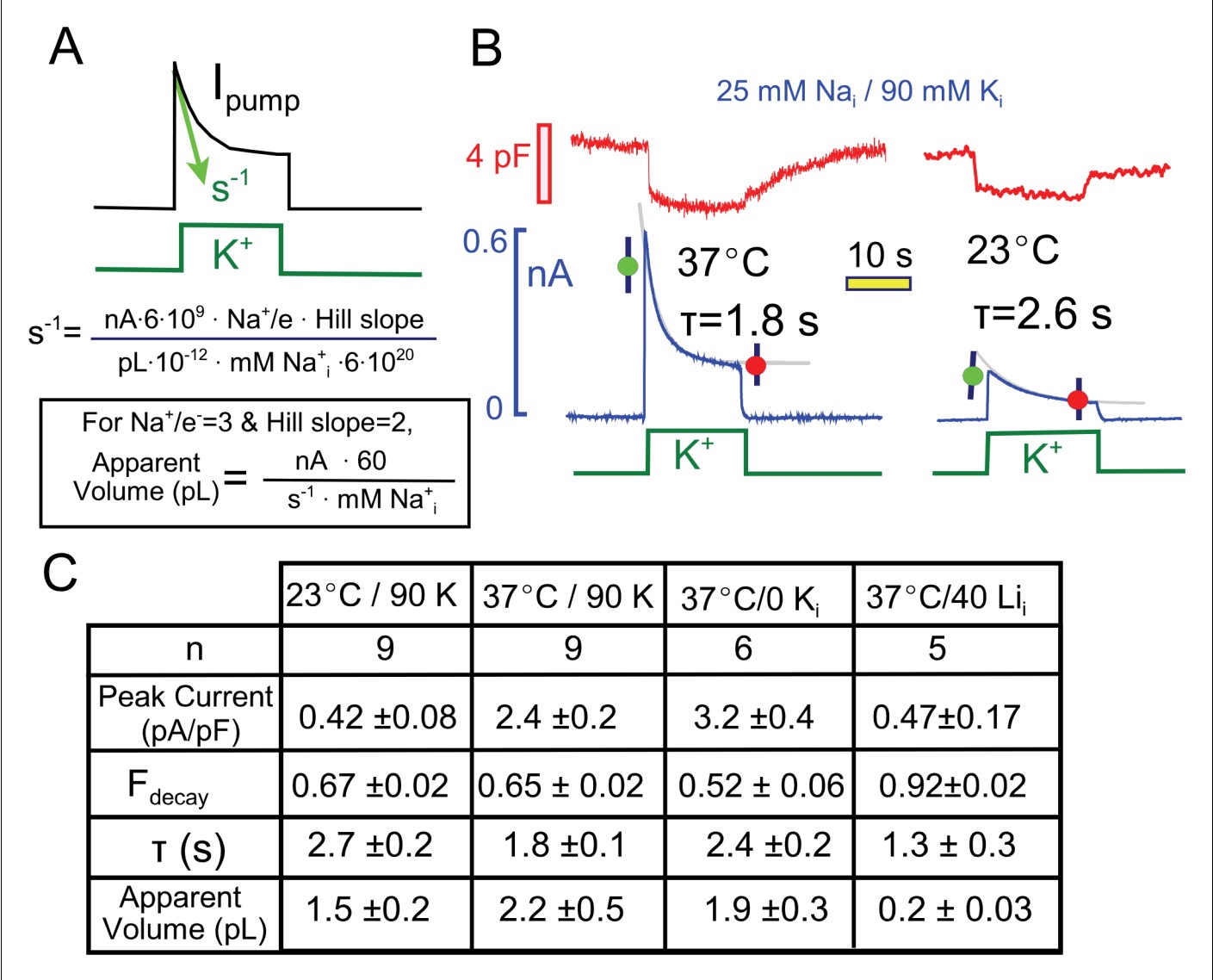

**Figure 3.** Quantitative analysis of pump current decay in relation to predictions for cytoplasmic Na depletion. (**A**) The initial rate of decline of current should be predictable from the current magnitude, the number of Na moved per charge moved (Na/e = 3), the slope of the Na-current relation ('Hill slope', ~2), the cytoplasmic volume in which Na mixes, and the cytoplasmic Na concentration. Alternatively, the apparent cell volume can be calculated from the other equation parameters. (**B**) Pump currents at 37 and 23°C with 25 mM Na and 90 mM cytoplasmic K with current decay fitted to single exponential functions. (**C**) Composite results from experiments with 25 mM cytoplasmic Na at 37°C and 23°C with 90 mM K Na, at 37°C without K, and at 37°C with 40 mM Li and no K. Pump currents are four-fold smaller at 23°C, but the fractional decay is similar; currents are increased by removal of K, but fractional current decay is decreased; and currents are small with a high Li concentrations, while fractional current decay is increased. All results are inconsistent simple depletion models.

37°C with four-fold larger current. The fractional decay of pump currents, plotted in *Figure 2F*, increases monotonically with decreasing Na concentration and is reasonably described by a simple reciprocal hyperbola, $K_d/(K_d+Na_i)$, with a $K_d$ of 12 mM. Instead of presenting Hill plots for this data set, we plot in both *Figure 2E and F* predictions from overtly simplified simulations of a depletion model (dotted lines) and an inactivation model (solid lines) that are described in Materials and methods. Clearly, the two postulates can account for these functional details equally well.

*Figure 3* presents multiple results that are not readily consistent with a Na depletion model, and *Figure 3A* highlights the major variables that would determine Na depletion in a restricted diffusion

model. The initial rate of pump current decline ($s^{-1}$) will depend on the number of Na ions extruded (i.e. the current magnitude in nA • $6 • 10^9 times$ **3**, the number of Na ions that move per elementary charge moved), the slope of the $Na_i$ concentration-current relationship (~2), the volume of the cytoplasm in which Na mixes, and the initial concentration of cytoplasmic Na. Rearranging the equation, one can calculate the apparent volume of cytoplasm in which Na is putatively depleting. As described next, values obtained for the cytoplasmic Na mixing volume are about 10% of the expected cytoplasmic volumes of murine myocytes (*Stagg et al., 2004*), and the apparent exchange times to this restricted volume are on the order of 2 to 3 s.

First, we performed a comparison of pump currents at 37° and 23°C with 25 mM Na and 90 mM K on the cytoplasmic side. *Figure 3B* presents individual experimental records together with their fits to single exponential functions. The fitted functions were used to calculate the fraction of current that decays ($F_{decay}$) and the apparent Na mixing volumes. As tabulated in *Figure 3C*, the peak Na/K pump currents were about five-fold smaller at 23°C than at 37°C. The fractional decay of current was however very similar (0.67 versus 0.65), while the decay time constant was larger and the apparent Na mixing volume was smaller at the lower temperature.

Also tabulated in *Figure 3C* are two further results that seem inconsistent with the Na depletion hypothesis. First, results for pump currents in the absence of cytoplasmic K are compared to results with cytoplasmic K. With 25 mM cytoplasmic Na, peak currents are increased by about 30%, whereas the fractional decay of current is decreased from 0.65 to 0.52. Second, the final column of the table in *Figure 3C* presents results of experiments using cytoplasmic lithium (Li) as a low affinity surrogate for cytoplasmic Na (*Hemsworth et al., 1997*; *Hermans et al., 1997*). With 40 mM Li in the pipette solution (30 mM TEA, 70 mM NMG, and no K), small pump currents (<0.5 pA/pF) could be activated by application of extracellular K. These currents showed unusually strong decay (>90%) in spite of their small magnitude, such that the apparent mixing volume of Li would be substantially less than a picoliter. This result is consistent with the inactivation model because more binding sites will be empty with a low affinity pump ligand, thereby promoting inactivation. In many other cases, we have observed that the fractional decay of pump currents increases, rather than decreases, as currents become smaller, and several clear examples will be considered with data presented in subsequent figures.

The kinetics of current decay are an additional source of information that can help distinguish between models. In equivalent experiments using rat myocytes, Verdonck et al. described that as cytoplasmic Na is increased from 0 to 100 mM, the rate constant that best describes pump current decay increases to a maximum with nearly the same Na dependence as pump current itself (*Verdonck et al., 2004*). This result seems inconsistent with the Na depletion model. If current decay were occurring as a result of Na depletion, the rate constant should have decreased as Na was increased above the half-maximal Na concentration of 8 mM and up to 100 mM. Furthermore, the fractional decay of current should have decreased markedly at the higher Na concentrations. Rather, the result supports an inactivation model in which the Na binding to inactive states promote recovery from inactivation. The kinetic results from our experiments were somewhat more complex: In the experiments described in *Figure 2* at 23°C, the initial rate of pump current decay *increased* as cytoplasmic Na was *decreased* from 0.06 ± 0.02 $s^{-1}$ at 40 mM Na to 0.14 ± 0.02 $s^{-1}$ at 2.5 mM Na. This favors our suggestion that inactivation occurs preferentially from E1 states that are not fully occupied by Na. Nevertheless, as evident subsequently in Figure 6B at 37°C, current decay can also accelerate as the fractional current decay becomes smaller at high cytoplasmic Na concentrations.

One specific prediction for the inactivation model was suggested by our previous work on Na-dependent inactivation of Na/Ca exchangers: The presence of *either* extracellular Na *or* Ca promotes exchanger transport reactions that move transporters from the equivalent E2 configurations to E1 configurations by the inward transport of bound ions. In other words, both extracellular Na and Ca promote accumulation of E1 states and favor inactivation (*Matsuoka and Hilgemann, 1994*). When experiments are performed in the presence of cytoplasmic Na and in the absence of extracellular Ca and Na, transporters initially accumulate in the E2 configuration that does not allow inactivation. Then, when Ca is applied to activate reverse transport (i.e. Ca influx), transporters shift to the E1 configuration and inactivation is promoted. When extracellular Na is applied *before* applying Ca, large fractions of the exchangers shift to the E1 configuration and inactivate without transporting Ca. That inactivation then becomes evident when Ca is applied without Na. For Na/K pumps the equivalent experiments will not normally be possible because the *reverse* pump reactions, which

move bound Na within E2 to E1 configurations, require that for cytoplasmic Na release the pump is dephosphorylated and that ATP is synthesized. These reverse reactions require, in addition to extracellular Na, high cytoplasmic concentrations of ADP. As described in *Figure 4*, appropriate experiments are readily possible to test this prediction by selecting conditions that promote cytoplasmic accumulation of ADP.

In describing the effects of extracellular Na on Na/K pump currents, we document first that just 5 mM Na in the extracellular solution without K suppresses for the most part an outward current that is very likely activated by contaminating K in nominally Na-free solutions, similar to results for squid axons (*Rakowski et al., 1989*). At the start of the experimental record, pump current was activated twice by moving the myocyte from a solution with 5 mM Na to one with 5 mM K. When 5 mM Na is replaced with 5 mM NMG, a small outward current develops that decays partially over 20 s, as expected for a pump current, and application of 5 mM K then activates a maximal pump current that is less than half of the previous pump currents. Return to a solution with 5 mM Na restores the pump currents. Application of 15 mM Na enhances pump currents only about 10% beyond peak currents obtained after incubation with 5 mM Na. We conclude therefore that 5 mM Na suppresses most of the pump activity supported by residual K in these solutions. As indicated within *Figure 4A*, the half-maximal Na concentration($K_d$) for this pump current rescue was 2.3 ± 0.3 mM (n = 6). This was determined in each experiment by designating the peak current in Na-free solution as the baseline for a rectangular hyperbola whose $K_d$ and maximum were then determined by peak currents at 2 higher Na concentrations. We also tested in 6 experiments the ability of tetraethylammonium (TEA) to rescue pump current, since TEA binds to E2 potassium sites without being transported (*Peluffo et al., 2004*). As expected, 120 mM TEA rescued pump activity equally well as 120 mM Na. However, much higher TEA concentrations were required than Na concentrations. The effect of 20 mM was very small, and the concentration dependence was concave upward with no evidence of saturation.

*Figure 4B* illustrates the protocol used to test whether extracellular Na (120 mM) can force pumps to inactivate. Equivalent to the experiment of *Figure 4A*, we first activated pump current as usual by replacing 6 mM Na with 6 mM K. Then, we switched to 120 mM Na (instead of NMG) for 40 s and reactivated the pump current with 6 mM extracellular K. The potentiation of current by 120 mM Na versus 7 mM amounted to 13 ± 3%, a change that is in good agreement with the $K_d$ of 2.3 mM. Thus, the activation of pump currents that occurs with increasing extracellular Na (*Garcia et al., 2013*) can be accounted for entirely by Na competition with contaminating K, thereby rescuing pumps from steady state inactivation..

*Figure 4C* illustrates, using the same protocol, that pump inactivation can be strongly promoted by extracellular Na under conditions that promote generation of cytoplasmic ADP, thereby allowing the reverse pump reactions to occur to the E1 state. Initially, we simply included 5 mM ADP in the pipette solution, together with 5 mM ATP. Peak pump currents then decreased significantly, but modestly, when extracellular Na was applied before pump current activation, as in *Figure 4B*. In this connection, we report that changes of nucleotides (e.g. use of 0.5 mM ATP versus 12 mM) in the pipette solutions in general had very little influence on pump currents, and in fact pump currents remained stable for periods of minutes with no added ATP. Clearly, ATP regenerating systems are very powerful in myocytes, and we therefore considered other means to decrease the ATP/ADP ratio. We included 10 mM deoxyglucose in the pipette solution with the expectation that hexokinases would rapidly phosphorylate deoxyglucose and thereby increase cytoplasmic ADP (*Russell et al., 1992*). As apparent in *Figure 4C*, Na/K pump currents then ran down progressively over 5 min. After substitution of 120 mM extracellular NMG by 120 mM Na for 40 s, the peak pump currents activated by K in the absence of extracellular Na were decreased by 35% on average. Furthermore, currents activated during application of extracellular K showed very little decay, as expected if inactivation had already occurred in the presence of extracellular Na. We report finally that similar results were often obtained *without* deoxyglucose when myocytes were employed more than 4 hr after isolation, consistent with a time-dependent run-down of ATP-regenerating systems.

## PKA and PKC activation have little or no effect on pump currents

From studies outlined in the Introduction, we expected that the 'standard' conditions of these experiments would be optimal to observe Na/K pump regulation by PKAs, and *Figure 5* presents results for activation of both PKAs and PKCs. Pump currents were activated, as usual, by replacing

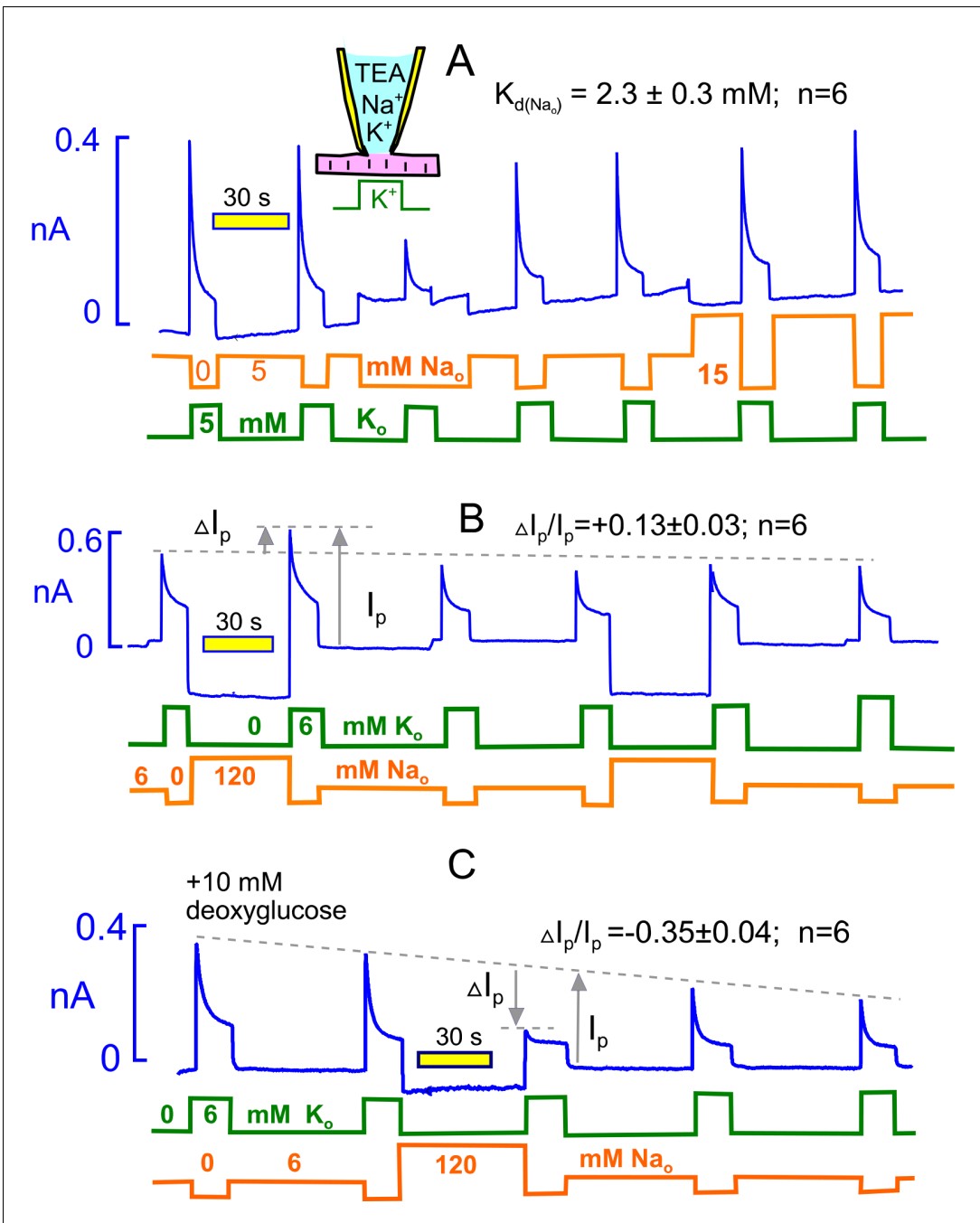

**Figure 4.** Extracellular Na rescues Na/K pumps from steady state inactivation but promotes inactivation when reverse transport reactions are enabled by ADP. (**A**) Employing standard solutions, pump currents were initially activated twice by replacing 5 mM Na with 5 mM K. Removal of Na without application of K activates a small current, and the pump current activated subsequently with 5 mM K is less than one-half of the control pump current. Thus, pumps inactivate in nominally Na free solution, presumably because pump activity supported by contaminating K is adequate to allow inactivation. Pump currents recover in the presence of 5 mM Na, and the application of 15 mM has very little further rescue effect. From 6 experiments, employing 5, 15 and 120 mM Na, the half-maximal Na concentration required to rescue pump currents was 2.3 mM. (**B**) Under standard conditions, 120 mM Na was applied, instead of NMG, for 30 s before applying K in the absence of Na. Peak pump currents were increased by 13% on average, compared to currents activated after incubation with 7 mM Na. (**B**) Na/K pump inactivation is promoted by conditions that allow backward pump reactions from E2 to E1 states. With 10 mM deoxyglucose in the pipette solution, to promote ATP hydrolysis and generation of ADP by hexokinsaes, pump currents ran down slowly over several minutes. After application of 120 mM Na for 30 s, peak pump currents

*Figure 4 continued on next page*

*Figure 4 continued*

activated by K were decreased to nearly their 15 s value, as expected if inactivation had occurred in the presence of extracellular Na. This result verifies a key prediction of the inactivation model presented in *Figure 1*.

7 mM Na for 7 mM K on the extracellular side in the presence of 25 mM Na and 90 mM K on the cytoplasmic side. As shown in *Figure 5A*, currents decayed by ~70% with a time constant of ~ 2 s. As evident in the second half of the recording, the β-adrenergic agonist, isoproterenol (0.5 µM), applied by shifting the myocyte to two separate solution lines containing the drug, caused no evident change of the activation-decay pattern or current magnitudes. *Figure 5B* shows composite results for isoproterenol in individual myocytes from 5 different myocyte preparations. Similar to results for isoproterenol, we also did not observe significant effects of the adenylate cyclase activator, forskolin (20 µM). Furthermore, no significant changes of the current decay and run-down patterns were observed when cAMP (2 mM) was included in the pipette solution. These results reiterate negative results using human iCell myocytes (*Fine et al., 2013*). In contrast to results for PKAs, we did observe small stimulatory effects of the PKC activator, rac-1-oleyl-2-acetyl- glycerol (OAG, 5 µM) on peak pump currents (~25%), similar to results of others (*Han et al., 2010*). These effects are however small in comparison to stimulatory effects of transient Ca elevations described next.

## Transient Ca elevations activate Na/K pump currents by increasing the apparent affinity for cytoplasmic Na

The weak Ca buffer power of our standard cytoplasmic solution (0.5 mM EGTA) allows myocytes to readily initiate spontaneous contractions via Ca-activated Ca release. To do so, we activate Ca influx via reverse Na/Ca exchange current for 3 to 5 s by applying Ca (5 mM) in the extracellular solution and then deactivate the current by returning myocytes to extracellular solution without Ca and containing 0.5 mM ETGA. Myocytes initiate a series of rapid contractions in this protocol, documented subsequently to involve Ca waves. Typically, contractile activity continues for a few seconds after exchange currents are deactivated. Contractions then stop and myocytes remain fully extended thereafter. *Figure 6A* illustrates the prominent after-effects transient Ca elevations when cytoplasmic Na is submaximal (25 mM with 90 mM K). During 15 to 40 s after the Ca elvation, membrane capacitance routinely increased by 3 to 6%. Within this same time frame, peak pump currents activated by extracellular K increased on average by 54%, and current decay during a 12 s K pulse was decreased from 70% to about 20%. Pump currents at 10 s were typically increased by three-fold. Over the course of 3 min, the Na/K pump currents returned to their initial wave form. Bar graphs to the right of the experimental record quantify representative results for 12 experiments, each using a myocyte from a different myocyte batch. We point out that these effects are the largest modulatory effects observed routinely on Na/K pump currents in the experience of this group. Further, we report that the myosin ii inhibitor, blebbistatin, at a high concentration (5 µM) blocked the observed contractile activity but did not block these effects (see *Table 1*). Therefore, the stimulatory effects must be caused by Ca elevations per se rather than mechanical activity.

*Figure 6B* shows results for the equivalent experiments when cytoplasmic Na was nearly saturating for the activation of Na/K pump currents (40 mM with 90 mM K). Pump currents still routinely displayed a rapid but small decay phase, which was fully ablated after the Ca elevation. Bar graphs to the right of the experimental record give composite results for 8 similar experiments. We point out that the Ca elevation caused a small decrease of peak pump currents, on average, and we describe subsequently much larger inhibitory effects of repeated Ca elevations on pump currents. In summary, the results of *Figure 6* document that the overall effect of transient Ca elevations is to increase the apparent affinity of Na/K pumps for cytoplasmic Na.

To explain the effects of Ca elevation in terms of a Na depletion model, one might propose that Ca elevation enlarges the subsarcolemmal Na space *or* enhances the diffusion of Na through the restricted space. We demonstrate in *Figure 7*, however, that much larger sustained Na fluxes can be generated by Na/Ca exchangers than are carried by Na pumps. For these experiments, we employed myocytes that overexpress NCX1 Na/Ca exchangers by at least five-fold in relation to WT expression levels (*Lin et al., 2013*) and that generate several nanoamperes of reverse exchange current in equivalent experiments. To limit the magnitude of Ca changes and to avoid excessive

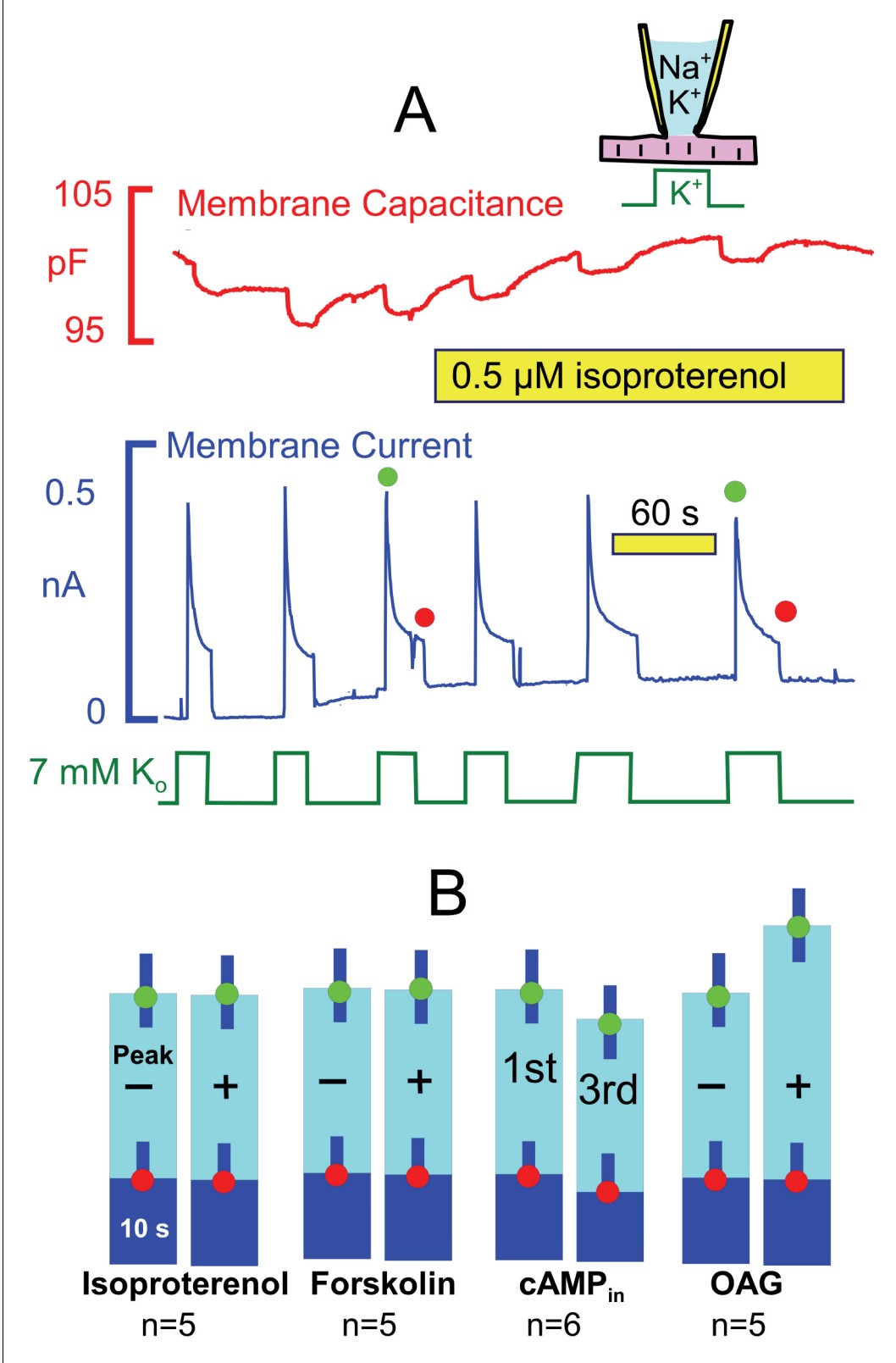

**Figure 5.** Stimulation of Na/K pump currents by activating protein kinase A (PKA) and protein kinase C (PKC) is weak or absent. (**A**) Under standard conditions, Na/K pump currents were activated repeatedly by applying extracellular K, and then isoproterenol (0.5 μM) was applied by moving the myocyte to separate solution lines containing the drug. (**B**) From left to right, composite results are presented for equivalent experiments in which

*Figure 5 continued on next page*

*Figure 5 continued*

isoproterenol (0.5 µM) or forskolin (20 µM) was applied to activate PKAs, in which 2 mM cAMP was included in pipette solutions, and in which 1-oleoyl-2-acetyl-sn-glycerol (OAG, 5 µM) was applied to activate PKCs. Only OAG has a significant effect to weakly increase the peak Na/K pump currents.

contraction, we employed heavily Ca- and pH-buffered pipette solutions without K (50 mM EGTA, 12 mM Ca, 50 mM HEPES, and 20 mM Na; resting free Ca, ~0.2 µM). Peak Na/K pump currents in these myocytes were not different in magnitude from WT myocytes (~2 pA/pF with 20 mM Na), but they decayed nearly completely. Peak exchange currents amounted to >20 pA/pF, and they decayed to a magnitude of 430 ± 33 pA after 20 s. Impressively, the maintained exchange current magnitudes approximate the maximal Na flux that can be expected for an access resistance of 3 MΩ and a pipette Na concentration that amounts to 7% of all diffusible ions (i.e. 20 mM out of 300 mM with ion fluxes calculated from a constant field equation and assuming a 3Na/1Ca exchange stoichiometry). The total magnitude of Ca influx during each episode of exchange current amounts to more than 10 mmol/Liter of cytoplasmic space, assuming a cell volume of 30 pL. In response to these Ca elevations, peak Na/K pump currents doubled in magnitude and the 10 s current magnitudes increased from negligible values to 76 ± 6 pA (n = 6). Clearly, it becomes difficult to explain these current patterns in terms of a restricted subsarcolemmal Na space shared by Na/K pumps and Na/Ca exchangers. The total amount of Na moved during the initial Na pump current over 20 s is subseqeuntly moved by Na/Ca exchangers within just a few milliseconds. Therefore, one must assume that the Na space available to Na pumps is very small (<1 pL), while that available to Na/Ca exchangers is very larger. Either the two transporters do not share the same Na space, or else Ca influx dramatically expands the space within milliseconds.

## Repeated Ca elevations can reactivate *or* strongly inhibit Na/K pump currents

*Figure 8A* presents optical measurements of cytoplasmic Ca changes that occur during two consecutive Ca elevation protocols using the protocol and conditions of *Figure 6*. For these measurements myocytes were loaded with the AM ester of Fluor4 (2 µM) (*Gee et al., 2000*) for 30 min. As necessary for optical measurements, the recordings were performed using mycoytes attached to cover slips, and solutions were changed with a device in which two solution lines shared a common volume outflow. Therefore, solution changes were not as abrupt as in routine experiments. Perhaps for this reason, or as a result of dye loading, myocytes did not visually contract as vigorously as in routine experiments. Nevertheless, the overall patterns were very similar. The whole-cell fluorescence is presented as the middle record in *Figure 8A*, and the upper trace is a time-expanded record for a cross-sectional optical slice taken roughly in the middle of the myocyte. The cross-section record documents the occurrence of profuse Ca waves which follow closely our routine observations of contraction. Ca waves become increasingly pronounced and rapid as the Ca signal declines. Thereafter, Ca waves decrease in magnitude and eventually in frequency. During application of extracellular Ca, the Fluor4 fluorescence increased on average 3.7 ± 0.4 fold (n = 7). This corresponds to a peak free Ca of 2.7 µM, calculated with $K_d$ for Ca of 300 nM and a background free Ca of 0.13 µM. Free Ca decays with a time constant of ~15 s. Importantly, the cytoplasmic Ca changes can be reproduced with good precision after 3 min. In this experiment the first and second exchange currents were similar in magnitude, but we describe next that exchange currents can change substantially. We conclude from these measurements that during the routine Ca elevation protocols free Ca rises to at least 2 and probably 3 µM, and that Ca returns nearly to baseline within 15 to 30 s.

*Figure 8B* illustrates that pump currents could usually be enhanced further by a second Ca elevation. As illustrated further in *Figure 8B*, Na/Ca exchange currents were often larger during the second application of Ca, and they activated immediately rather than with a multi-second time course. Importantly, the enhanced Ca-mediated current turned off immediately upon removal of extracellular Ca when the application period was brief. This behavior indicates that the current depends directly on extracellular Ca and therefore is indeed Na/Ca exchange current, rather than a cytoplasmic Ca-activated nonspecific current that would turn off with the slower time course of the

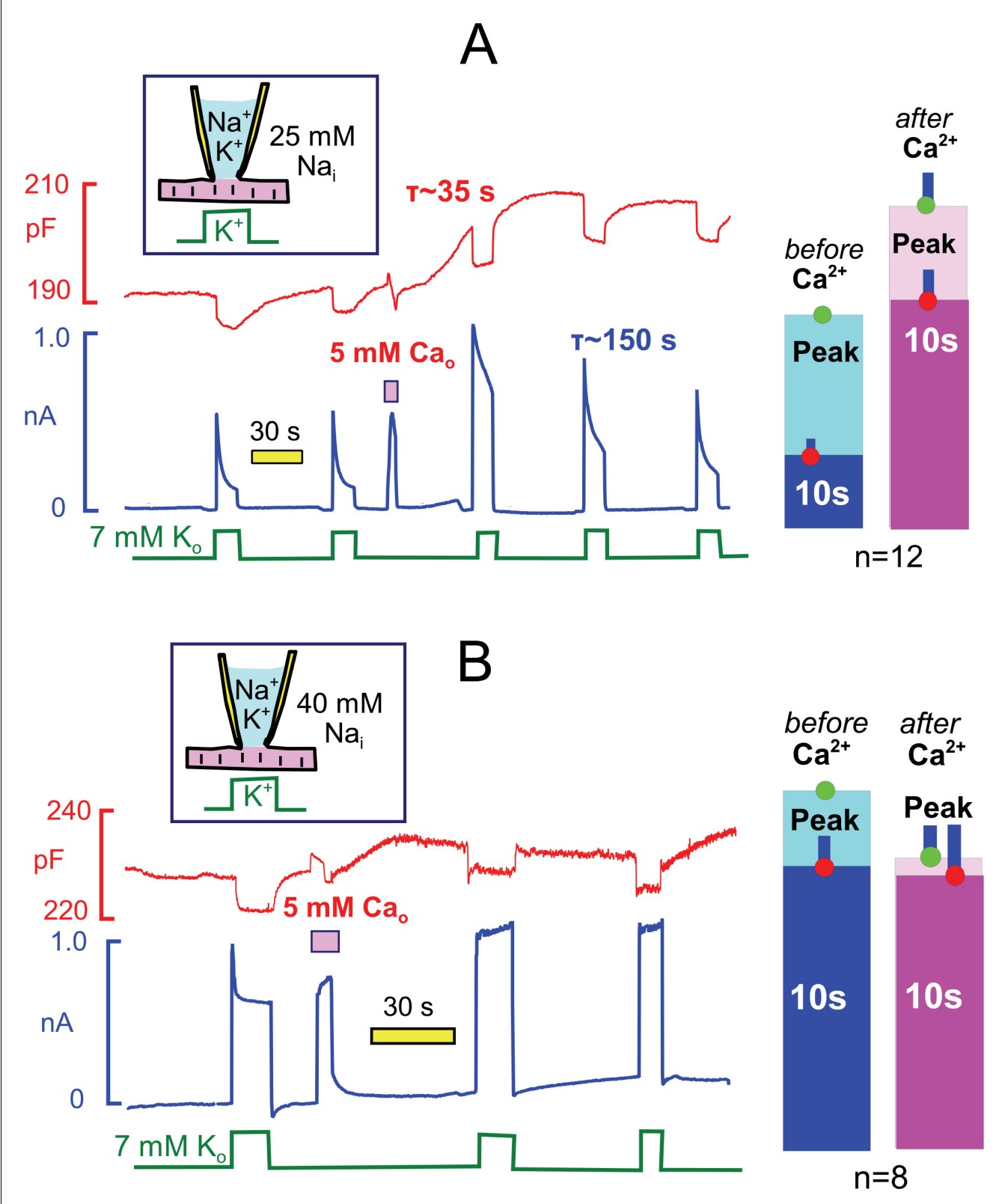

**Figure 6.** Transient Ca elevations robustly activate Na/K pump currents when cytoplasmic Na is submaximal but do not enhance maximal pump currents (**A** versus **B**). Na/K pump currents were activated repeatedly under standard conditions with 25 mM Na$_i$ (**A**) and with 40 mM Na$_i$ (**B**) for 15 s. Between two activation sequences, reverse Na/Ca exchange current was activated for 3 s by application of Ca (5 mM). After a few vigorous contractions, myocytes relaxed completely and then remained relaxed, while membrane capacitance typically increased by 2 to 5% over 30 s. With 25

*Figure 6 continued on next page*

*Figure 6 continued*

mM Na$_i$ (**A**), peak pump currents often increased by two-fold with an average of 54%, current decay during application of K was largely attenuated, and pump current at 10 s was increased three-fold on average. Thereafter, pump currents decreased nearly to baseline values with a time constant of ~150 s. With 40 mM Na$_i$(**B**), pump currents decayed rapidly by 19% on average, and this decay was fully attenuated after the Ca elevation. Peak currents were decreased by 18% on average after the Ca elevation. Average changes of peak and 10 s pump currents are given in bar graphs to the right of both recordings.

cytoplasmic Ca decline. In conclusion, Ca-dependent mechanisms enhance both Na/K pump and Na/Ca exchange activities for periods of more than one minute after a transient Ca elevation. Potentially, therefore, the same mechanism might modulate both currents.

It was important in these experiments to use rather short Ca influx episodes (3 to 5 s) because longer Ca elevations, or repeated Ca elevations, caused pump currents to rapidly run down. When Ca elevations were repeated a second, third, and fourth time, pump currents decreased in magnitude with increasing likelihood and often became negligible. As illustrated in the record in *Figure 8C*, the outward Na/Ca exchange currents ran down in close parallel to pump currents. This parallel run-down was usually associated with a fall of membrane capacitance, reflecting Ca-activated endocytosis (*Lariccia et al., 2011*). But even when the loss of membrane area was small, about 10% in *Figure 8C*, pump currents were strongly suppressed, usually ablated. Thus, transient Ca elevations have both facilitating and inhibiting effects on both Na/K pump currents and Na/Ca exchange currents. While the stimulatory effects invariably reversed over 3 min, reversal of inhibitory effects of repeated Ca elevations was highly variable.

Since Na/Ca exchange currents change in parallel with pump currents in response to Ca elevation, we next tested whether Ca influx by mechanisms beside Na/Ca exchange could activate Na/K pump currents. *Figure 9* demonstrates that Ca influx through Ca channels can indeed activate Na/K pump currents at a physiological cytoplasmic Na concentration (10 mM), and that cytoplasmic K can be replaced by cesium (Cs) in the protocol. To develop a large Ca current, we applied the Ca channel agonist FPL64176 (5 μM) (*Baxter et al., 1993*) throughout the experiment. Application of 6 mM extracellular Ca then activated a robust Ca current that decayed within a few seconds. Thereafter, both peak and 15 s Na/K pump currents were increased by >50% in 5 similar observations.

## Stimulatory effects of Ca elevations do not appear to involve classical signaling pathways

Next, we attempted to determine by what molecular mechanism(s) Ca elevations enhance pump currents. *Table 1* summarizes the extensive range of these experiments using the standard conditions and protocol of *Figure 6*. All results are normalized to the peak current that occurred when pump currents were activated just before Ca was applied. From left to right, the columns give initial peak current (100%), the relative current magnitude after 10 s, and the fractional decay of current before the Ca elevation. These values are followed by the equivalent *post-Ca* values for peak current, current after 10 s, and the fractional decay at 10 s of the first pump current transient after the Ca elevation. The *Control* data (25 mM Na$_i$ with 90 mM K$_i$) are averages from 12 representative experiments from 12 different batches of myocytes. Unless indicated otherwise, test compounds were included in both cytoplasmic and extracellular solutions. In short, we did not clearly implicate any classical signaling mechanism in the stimulation of Na/K pump activity by Ca elevations.

### Summary of tabulated results from top to bottom

1. Basic mechanism of Ca action. The myosin ii inhibitor, blebbistatin, at 5 μM decreased the stimulatory effect of Ca elevations by 25%. Given the hydrophobic nature of this agent, the high concentration employed, and the small magnitude of the effect, the result does not seem significant. The SERCA Ca pump inhibitor, thapsigarin (2 μM), was without effect, and a high concentration of ryanodine (2 μM) had at most a small effect. Accordingly, Ca influx by Na/Ca exchange provides an adequate Ca signal to facilitate pump currents without internal Ca release.

2. Phospholemman. Using myocytes from mice lacking the regulatory protein, phospholemman, the fractional decay of current, both before and after Ca, was reduced in comparison to WT

**Table 1.** Normalized results for interventions tested to modify pump current stimulation by Ca elevations using the standard protocol (25 mM $Na_i$ with 90 mM $K_i$). Currents are normalized to the peak current before application of Ca. Results are given for peak and 10 s currents before applying extracellular Ca, followed by the average fractional decay of that current, and for the first peak and quasi steady state (10 s) current after applying Ca, followed by the average fractional decay of that current. See text for annotation of the major results. Abbreviations: PLM, phospholemman; BIM, Bisindolylmaleimide; DTT, dithiothreitol; SOD, Superoxide Dismutase.

| Pre-Ca2$^+$ | | | | Post-Ca2$^+$ | | | |
|---|---|---|---|---|---|---|---|
| | Peak | 10 s | $F_{decay}$ | Peak | 10 s | $F_{decay}$ | n |
| Control (25 mM Nai) | 100 | 35 ± 2 | 0.65 | 154 ± 12 | 111 ± 10 | 0.28 | 12 |
| **1. Basic Mechanism** | | | | | | | |
| Blebbistatin (5 µM) | 100 | 44 ± 8 | 0.60 | 138 ± 8 | 80 ± 15 | 0.40 | 6 |
| Thapsigargin (2 µM) | 100 | 36 ± 5 | 0.74 | 198 ± 70 | 118 ± 50 | 0.41 | 4 |
| Ryanodine (2 µM) | 100 | 34 ± 5 | 0.67 | 154 ± 8 | 84 ± 10 | 0.44 | 5 |
| **2. Phospholemman (PLM)** | | | | | | | |
| PLM -/-mice | 100 | 74 ± 4 | 0.26 | 125 ± 10 | 114 ± 10 | 0.09 | 9 |
| **3. Membrane cytoskeleton** | | | | | | | |
| Phalloidin, 3 µM | 100 | 57 ± 10 | 0.43 | 147 ± 25 | 116 ± 20 | 0.19 | 5 |
| Latrunculin, 3 µM | 100 | 37 ± 11 | 0.63 | 135 ± 9 | 96 ± 22 | 0.25 | 4 |
| **4. Serine/threonine phosphorylation** | | | | | | | |
| KN93, 3 µM | 100 | 54 ± 7 | 0.46 | 121 ± 13 | 112 ± 10 | 0.06 | 4 |
| CAMK Pep., 15 µM | 100 | 57 ± 6 | 0.43 | 131 ± 9 | 93 ± 9 | 0.28 | 6 |
| CK59, 20 µM | 100 | 25 ± 6 | 0.78 | 183 ± 30 | 75 ± 2 | 0.54 | 4 |
| H7, 0.3 mM | 100 | 35 ± 4 | 0.65 | 157 ± 12 | 95 ± 10 | 0.39 | 6 |
| BIM, 4 µM 1 hr | 100 | 27 ± 4 | 0.73 | 131 ± 4 | 86 ± 5 | 0.34 | 6 |
| PKC19-31, 10 µM | 100 | 24 ± 1 | 0.77 | 195 ± 58 | 136 ± 46 | 0.34 | 9 |
| Cyclosporin A, 3 µM | 100 | 48 ± 6 | 0.52 | 126 ± 6 | 101 ± 5 | 0.25 | 5 |
| **5. Redox and NOS signaling** | | | | | | | |
| PRD6 -/- mice | 100 | 33 ± 5 | 0.66 | 189 ± 24 | 134 ± 17 | 0.26 | 8 |
| Oxy-glutathione, 5 mM | 100 | 31 ± 4 | 0.69 | 195 ± 18 | 120 ± 11 | 0.37 | 6 |
| Glutathione, 8 mM | 100 | 38 ± 7 | 0.62 | 130 ± 14 | 89 ± 15 | 0.30 | 4 |
| DTT, 0.2 mM (in & out) | 100 | 28 ± 3 | 0.73 | 148 ± 31 | 100 ± 35 | 0.35 | 5 |
| SOD, 3 mg/ml | 100 | 26 ± 2 | 0.74 | 170 ± 18 | 101 ± 12 | 0.67 | 5 |
| L-NAME, 0.1 mM | 100 | 25 ± 6 | 0.75 | 190 ± 41 | 120 ± 27 | 0.36 | 4 |
| **6. Mitochondrial signaling** | | | | | | | |
| CGP37157, 20 µM | 100 | 22 ± 4 | 0.78 | 184 ± 24 | 83 ± 15 | 0.55 | 4 |
| RU360, 20 µM | 100 | 44 ± 7 | 0.56 | 111 ± 7 | 94 ± 15 | 0.17 | 6 |
| **7. Lipid metabolism** | | | | | | | |
| PLD1-PLD2 -/-mice | 100 | 34 ± 3 | 0.66 | 114 ± 4 | 86 ± 9 | 0.25 | 9 |
| DHHC5-GT mice | 100 | 31 ± 3 | 0.69 | 115 ± 5 | 83 ± 5 | 0.27 | 6 |
| U73122, 10 µM | 100 | 39 ± 4 | 0.61 | 165 ± 8 | 117 ± 11 | 0.29 | 5 |
| Wortmannin, 10 µM | 100 | 48 ± 6 | 0.51 | 125 ± 8 | 120 ± 10 | 0.03 | 4 |
| **8. Nonspecific membrane modifiers** | | | | | | | |
| Genistein, 25 µM | 100 | 16 ± 6 | 0.84 | 6 ± 0 | 0 ± 0 | – | 5 |
| Daidzein, 20 µM | 100 | 40 ± 4 | 0.59 | 104 ± 9 | 54 ± 5 | 0.48 | 6 |
| Quercetin, 100 µM | 100 | 57 ± 11 | 0.43 | 96 ± 54 | 62 ± 31 | 0.31 | 5 |
| Methylene Blue, 3 µM | 100 | 41 ± 4 | 0.59 | 54 ± 19 | 42 ± 15 | 0.22 | 8 |
| Triacsin C, 2 µM | 100 | 43 ± 6 | 0.57 | 43 ± 21 | 31 ± 15 | 0.25 | 7 |

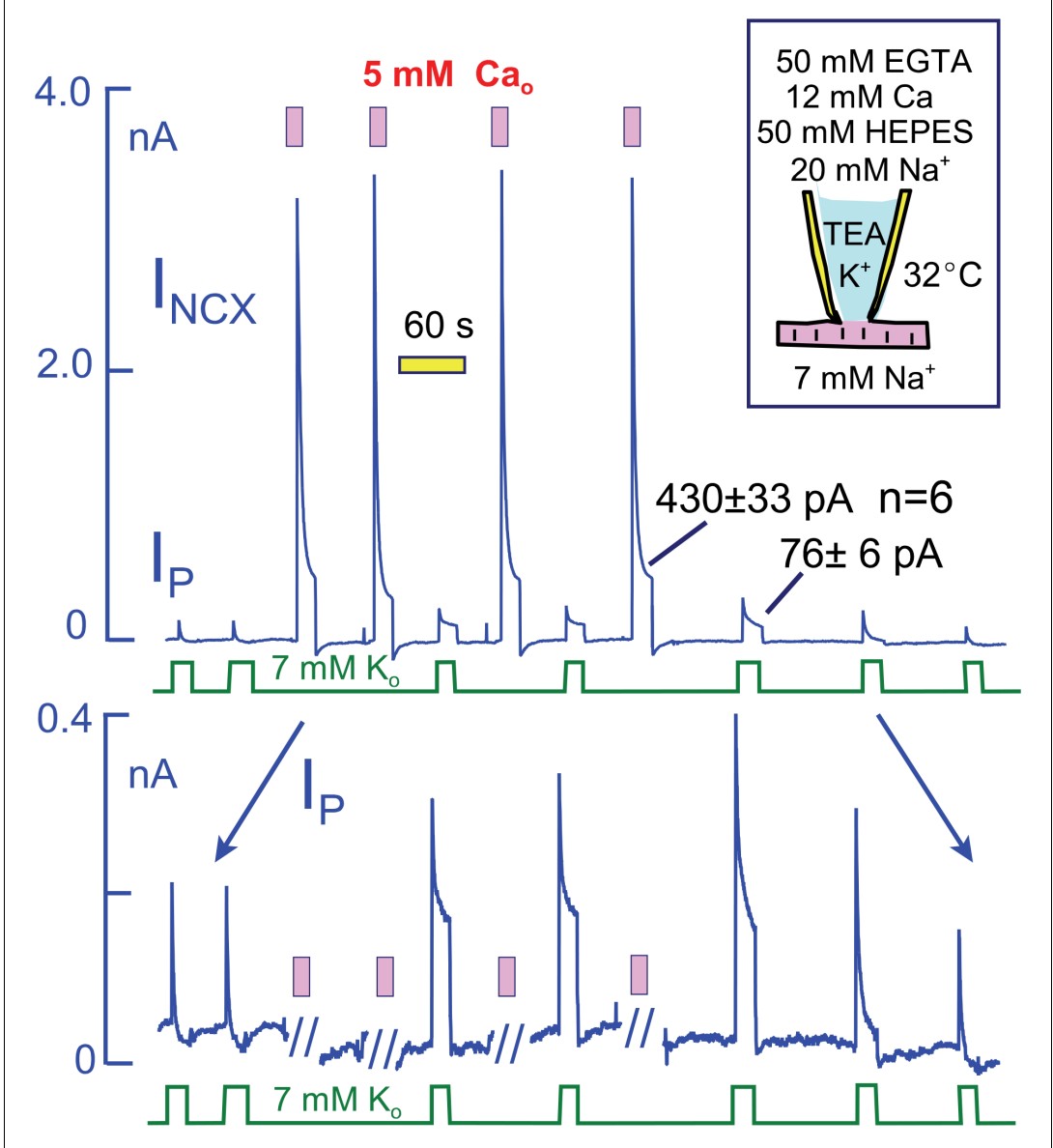

**Figure 7.** Na/K pump and Na/Ca exchange currents in myocytes that overexpress NCX1 Na/Ca exchangers by more than five-fold. The cytoplasmic (pipette) solution is designed to highly buffer free Ca and protons. The magnitudes of exchange currents after 15 s are much greater than the 15 s pump currents. Thus, Na/K pumps and Na/Ca exchangers cannot be sharing a restricted subsarcolemmal space. **Top panel**. Complete current record in which Na/K pump currents and Na/Ca exchange currerrents were activated alternately. **Bottom panel**. Expanded view of the current record to accurately depict the Na/K pump current. Pump currents decay completely before activation of Na/Ca exchange. In response to Na/Ca exchange currents, peak pump currents double and a maintained current develops. These changes reverse completely over 3 min after the last Ca elevation.

myocytes. Thus, current decay is partially dependent on phospholemman. However, Ca elevations reliably enhanced pump currents and decreased the fractional decay in phospholemman KO myocytes. We conclude therefore that phospholemman does not mediate the stimulatory effect of Ca elevations.

3. Membrane cytoskeleton.Na/K pumps are coupled to membrane cytoskeleton in intact cells (*Mohler et al., 2005*). Nevertheless, the standard actin reagents, latrunculin and phalloidin, at high concentrations (3 µM) did not clearly modify pump function when examined in the standard protocol after incubation for periods of at least 5 min before experiments.

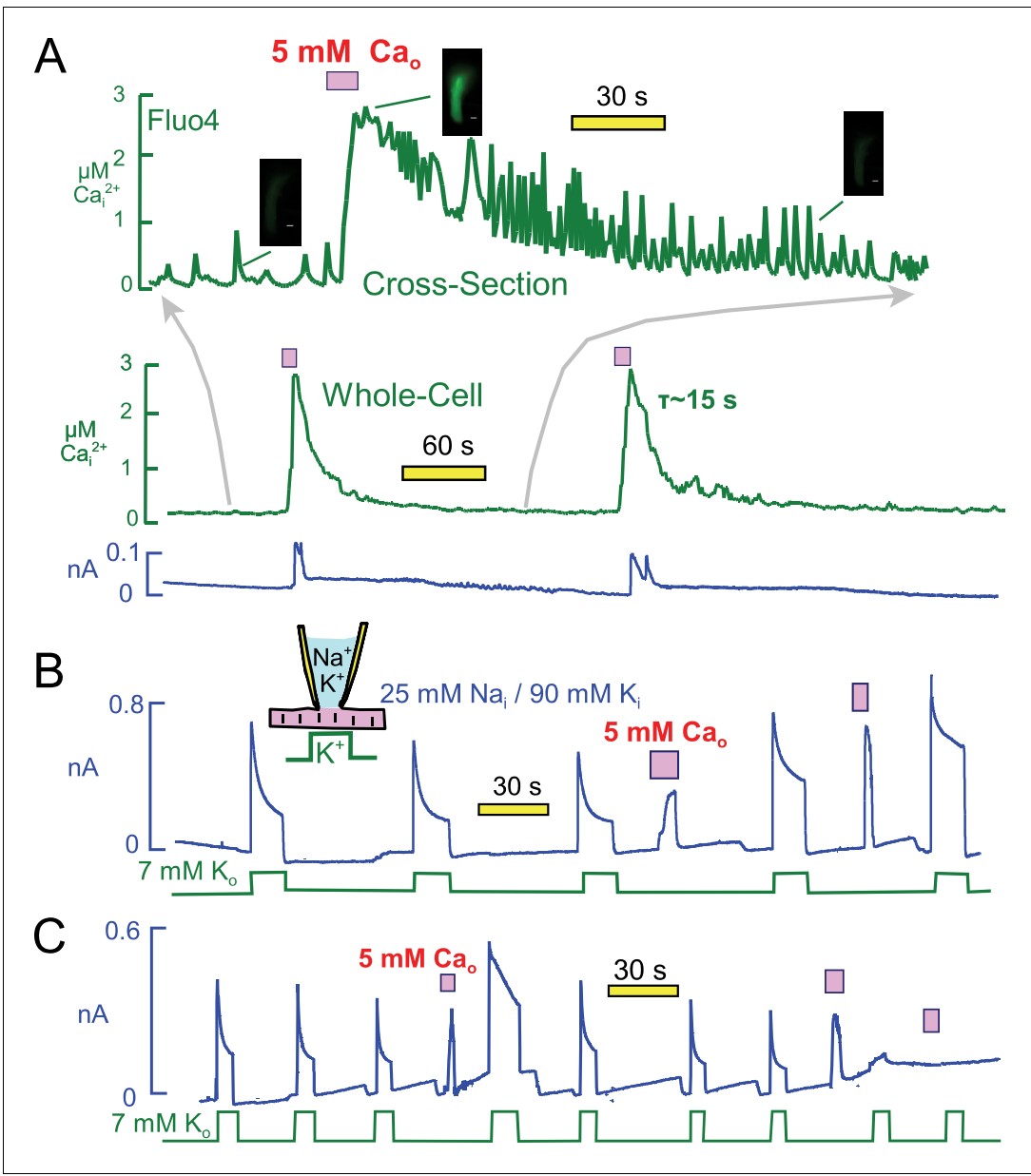

**Figure 8.** Free cytoplasmic Ca changes and their effects on Na transport currents during repeated Ca elevations. (**A**) Free Ca measurements with Fluo4. The center record presents the whole-cell free Ca signal measured with Fluo4. Fluorescence increased by 3.7 ± 0.4 fold (n = 7) during application of extracellular Ca and returned to baseline with a time constant of ~25 s. Correcting for dye nonlinearity, the fall of cytoplasmic Ca probably occurs with a time constant of ~15 s. The upper record shows in an expanded time scale fluorescence from a cross-sectional slice in the middle of themyocyte, revealing that extensive Ca waves are taking place in the myocytes. These waves become more rapid an large as free Ca declines, and finally they decline in both magnitude and frequency. (**B**) Representative for >400 observations, Na/Ca exchange current shows a time-dependent activation phase on first application of extracellular Ca. *This secondary rise corresponds to the regulatory activation of exchangers that occurs as cytoplasmic Ca rises (**Matsuoka et al., 1993**).* On second application of Ca, the exchange current activates immediately and to a higher magnitude than on initial Ca application. Thereafter, pump current is increased beyond its peak after the first Ca elevation. (**C**) After three or more Ca elevations, and *infrequently* after the second repetition, both Na/K pump and exchange currents decreased and often became negligible. The loss of transporter currents was often accompanied by declining membrane capacitance during Ca elevations, indicative of massive endocytosis (**Lariccia et al., 2011**).

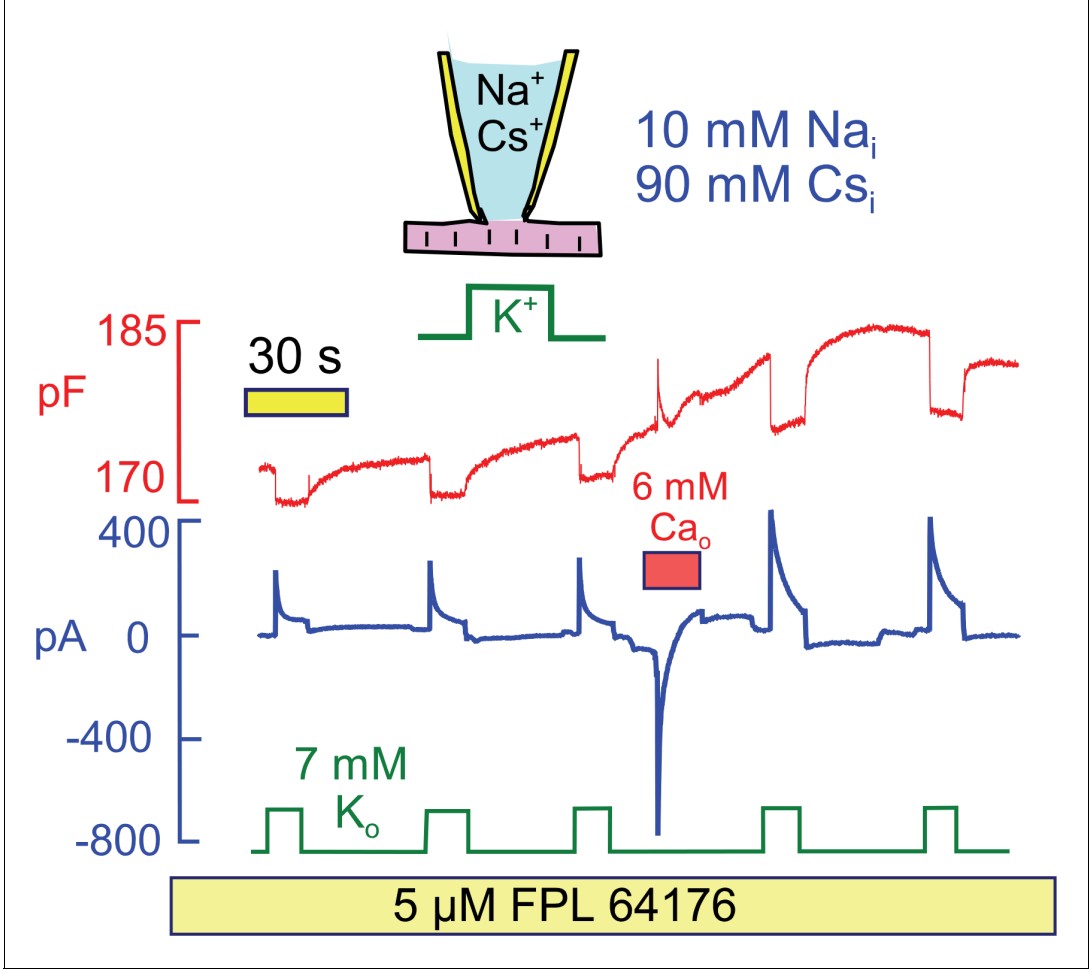

**Figure 9.** Stimulation of Na/K pump current in the presence of physiological (10 mM) cytoplasmic Na and in the absence of cytoplasmic K. To evoke significant Ca influx in the presence of low cytoplasmic Na, FPL64176 was employed to lock L-type Ca channels open at 0 mV. On application of 6 mM extracellular Ca, inward Ca current amounted several hundred pA, and current inactivated with a time constant of about 1 s. Thereafter, $C_m$ increased by 3%, similar to results with reverse Na/Ca exchange, and peak and 12 s pump currents were roughly doubled. Similar results were obtained in 4 similar experiments.

4. Serine/threonine phosphorylation.Three calmoulin-dependent (CAM) kinase inhibitors (KN93 (5 µM), a calmodulin-inhibitory peptide from CAM kinase (CAM-PEP, 20 uM; CaM Kinase II (290–309)), and CK59 (20 µM)) did not substantively modify pump function or effects of Ca. The non-selective serine protein kinase inhibitor, H7, which blocks stimulation of pump currents by PKA activation (*Gao et al., 1996*), was without effect at a high concentration (0.5 mM), as were two PKC reagents (bisindolylmaleimide (BIM), 3 uM, and the pseudosubstrate inhibitor, Kemptide (peptide 19–31, 10 uM)). The stimulatory effects of Ca elevations were furthermore unaffected by inhibitors of Ca-dependent phosphatases (cyclosporine A, 2 µM; FK506, 10 µM).

5. Redox and NOS signaling. In relation to proposed roles of ROS and glutathionylation in regulating Na/K pumps, we analyzed pump function in myocytes from Peroxiredoxin6 (PRDX6)-/- mice (*Nagy et al., 2006*). Pump function and responses to Ca elevations remained robust and comparable to WT myocytes. Neither oxidized nor reduced glutathione at high concentrations (5 and 8 mM) substantively modified pump function when included in the pipette solutions. DTT (0.2 mM) was without significant effect when employed on both membrane sides, and a very high concentration of superoxide dismutase (SOD, 3 mg/ml), included in the pipette solution, was without significant effect. A high concentration of the NO scavenger (L-NAME) was also without effect.

6. Mitochondrial involvement. As mentioned above, cyclosporine A was without effect, suggesting that permeability transition pores are not important. Na and Ca movements via the mitochondrial Na/Ca exchanger do not appear to be involved, as the mitochondrial Na/Ca exchange inhibitor CGP37157 was without effect. Also, an inhibitor of the mitochondrial Ca uniporter, RU360, was without effect at a high concentration (20 µM).

7. Lipid metabolism. Pump function was similar in mice lacking phospholipase D activity, i.e. in PLD1/PLD2 double knockouts (*Ali et al., 2013*) and in myocytes from mice lacking the surface membrane acyl transferase, DHHC5 (*Li et al., 2012*). Neither $PIP_2$ breakdown nor $PIP_2$ synthesis seems to be critical, as the phospholipase C inhibitor, U73122 (10 µM), and the PI4P/PI3P kinase inhibitor wortmannin (10 µM) were without effect.

8. Nonspecific membrane modifiers. The final five reagents in *Table 1* did substantively reduce the stimulatory effects of Ca elevations. All five reagents are used in signaling studies, but may be expected to have nonspecific actions on membranes (*Hwang et al., 2003*): genistein (25 µM), daidzein (20 µM), quercetin (100 µM), methylene blue (3 µM) and triascin C (2 µM). Of these five, genistein was notable in that it caused progressive current run-down with increasingly strong current decay during pump activation. Four reagents (genistein, quercetin, methylene blue and triascin C) promoted a phenotype in which application of Ca caused strong inhibition of pump currents on its first application. In the case of daidzein, commonly used as a control compound for tyrosine kinase inhibition by genistein (*Akiyama et al., 1987*), the stimulatory effects of Ca elevation were blocked without promoting an inhibitory effect of Ca. Therefore, we conclude that hydrophobic/amphipathic compounds can indeed block the stimulatory effects of Ca elevation in these experiments. The relative potency of all five reagents, together, suggests that cardiac Na/K pump activity is rather sensitive to nonspecific perturbation of the sarcolemma.

## Ca elevations and metabolic stress may act via physical-chemical changes of the sarcolemma

Given our failure to clearly implicate classical signaling mechanisms, we next asked whether transient Ca elevations might act by changing physical-chemical properties of the sarcolemma that influence pump activity. We point out in this connection that large transient Ca elevations cause the plasmalemma of multiple cell types to become disordered, such that numerous hydrophobic compounds bind to the cell surface more avidly (*Hilgemann and Fine, 2011*). As described in *Figure 1*, Ca elevations occurring in our standard protocol indeed enhance the binding of both hydrophobic cations and anions to the cardiac sarcolemma.

*Hexyltriphenylphosphonium (HTPP)* is a hydrophobic cation that crosses membrane bilayers in the absence of transporters, owing to its hydrophobicity and delocalization of its positive charge (see (*Hilgemann and Fine, 2011*) for further explanations). *Figure 10A* presents parallel measurements of Na/K pump currents and inward currents generated by HTTP (20 µM), which was applied repeatedly for 3 s during the standard protocol. HTTP current amounted to −180 pA initially. After the transient Ca elevation, the peak Na/K pump current was doubled, and the HTTP current was increased to −270 pA (i.e. by 50%). Thereafter, both the Na/K pump current and the HTTP current decreased toward baseline with similar 3 min time courses. Statistics for 5 experiments are given within *Figure 10A*. Notably, HTTP currents decreased below their initial values when Ca elevations were repeated and Na/K pump currents also decreased below baseline, as described in *Figure 8C*. Thus, both the stimulatory and inhibitory effects of Ca elevations are potentially occurring through changes of the membrane itself.

To test whether membrane changes caused by Ca transients might be electrostatic in nature, and therefore enhance specifically the binding of positively charged amphipaths, we examined capacitive signals that are detected when anionic amphipaths bind to cells and expand the membrane (*Fine et al., 2011*; *Hilgemann and Fine, 2011*). *Figure 10B* shows results for the hydrophobic anion and Cl channel blocker, niflumic acid (NFA). NFA causes a 1 to 3% increase of capacitance when applied at a concentration of 100 µM. This capacitive binding signal nearly doubled after transient Ca elevations that caused an approximate doubling of Na/K pump currents and a 5% increase of capacitance. Statistics for 5 similar experiments are given within *Figure 10B*. Given that both HTTP and NFA signals increase by at least 30 and 80%, respectively, while membrane capacitance increases by only 3 to 8%, we conclude that the membrane is becoming more disordered in response to Ca elevations.

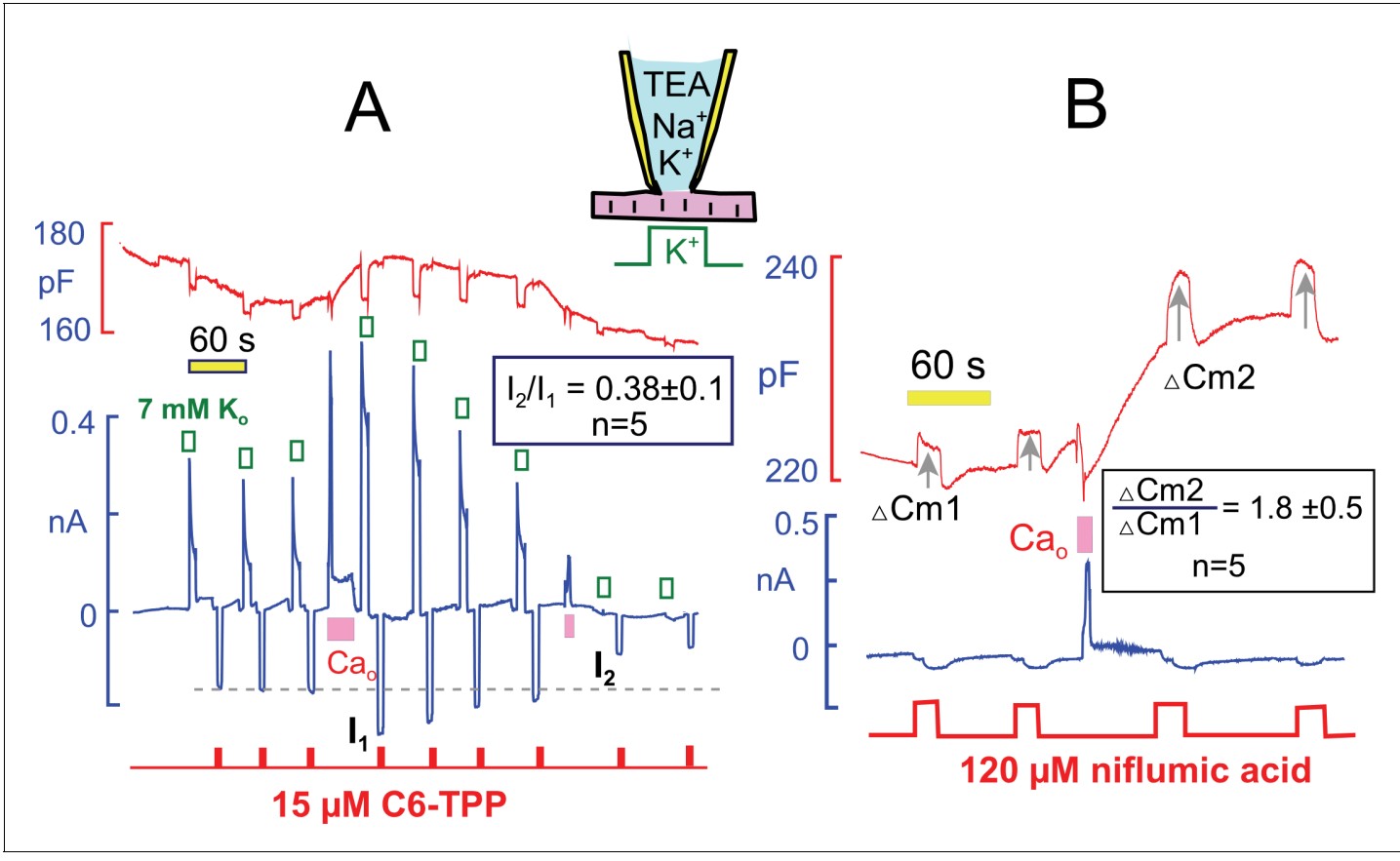

**Figure 10.** Interactions of hydrophobic cations and anions with the sarcolemma track the stimulation and inhibition of pump currents by Ca elevations. (A) Na/K pump currents were activated repeatedly for 8 s and hexyltriphenylphosphonium (C6-TPP, 15 µM) currents were activated alternately with the pump currents for 3 s. Pump current increased and then decreased in the usual manner and rate in response to Ca elevation. Currents carried by C6-TPP increased by 39% on average and then decreased in parallel with pump currents. A second Ca elevation causes profound inhibition of pump current and a 60% decrease of C6-TPP current. As indicated, C6-TPP currents decreased on average by 62% during the time over which pump currents ran down almost completely. (B) Niflumic acid induces a capacitive binding signal, when applied at a concentration of 120 µM, which propably involves expansion of the membrane. In response to Ca elevation, the capacitive binding signal increases on average by 80% (n = 5). Thus, both cationic and anionic membrane probes bind more effectively after Ca elevations.

Next, we examined in similar experiments whether the run-down of Na/K pump currents, that is observed when deoxyglucose is included in the cytoplasmic solution (see *Figure 4C*), also modifies C6-TPP currents. Using standard solutions (25 mM $Na_i$) with 10 mM deoxyglucose in *Figure 11*, pump currents ran down completely in the course of 10 min, and the fractional decay of pump current ($F_{decay}$) increased to essentially complete decay as currents decreased. We mention that pump currents did not run down significantly when 10 mM glucose-6-phosphate (n = 4) or 10 mM deoxyglucose-6-phosphate was included in pipette solutions (n = 3). Therefore, the rapid phosphorylation of deoxyglucose by hexokinases, at the expense of ATP, is likely to be the cause of run-down in these experiments, rather than mechanisms downstream of deoxyglucose-6-phosphate. As summarized in the Table in *Figure 11*, inward HTPP currents decreased in parallel with pump currents, on average by 69%. Furthermore, HTPP currents were stable when deoxyglucose was not included in the pipette, and HTPP currents decreased by 51% when the pipette solutions contained 140 mM K, instead of NMG, and no Na. Thus, the membrane changes that occur with deoxyglucose do not appear to depend on Na-dependent Na/K pump conformational changes. Rather, they appear to be caused by ATP/ADP changes, per se, exerting relatively rapid effects on proteins that in turn influence the bilayer.

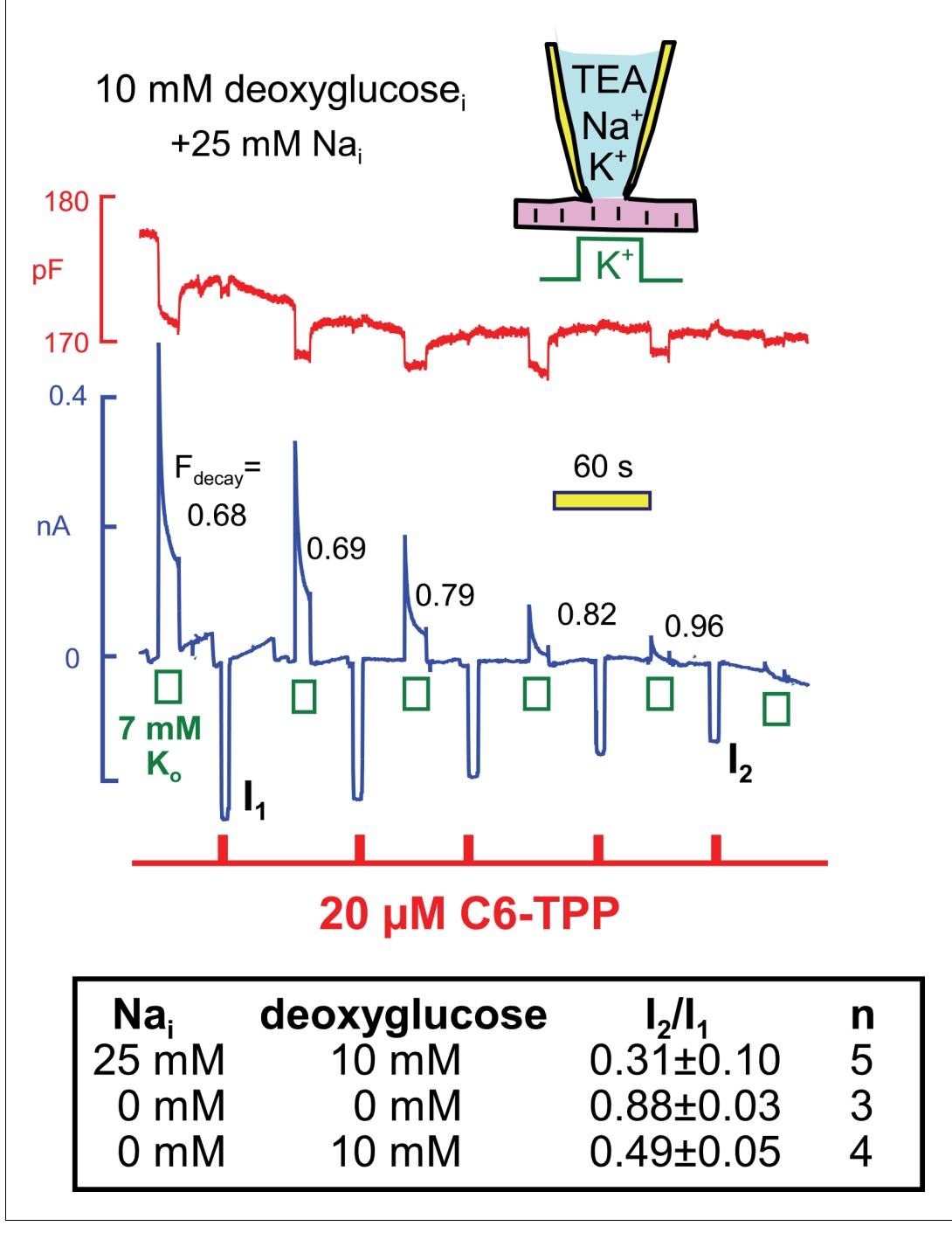

**Figure 11.** Parallel decay of pump currents and C6-TPP currents when deoxyglucose (10 mM) is included in the pipette solution. Pump currents were activated for 10 s alternatively with C6-TPP currents for 2 s. Fractional decay of pump currents increases from 0.68 to > 0.9 as pump current declines, and the C6-TPP current decreases by 69% on average. The table beneath the figure gives statistics for equivalent experiments *without* deoxyglucose, *with* deoxyglucose but *without* Na, and *with* deoxyglucose *and* 25 mM cytoplasmic Na.

## Hydrophobic cation currents track changes of Na/K pump and Na/Ca exchange activity

Hydrophobic fluorescent dyes have been used for decades to analyze electrogenic reactions of the Na/K pump (*Bühler et al., 1991*). The interpretation of those experiments suggests that ion binding within the pump exerts long-range effects on the intramembrane electrical field that are sensed by voltage-dependent dyes some distance away. Conformational changes of the Na/K pump per se may also affect these dyes (*Fedosova et al., 1995*). With this background, we tested whether conformational changes of Na/K pumps during pump current elevations might affect the currents carried by HTTP. As shown in *Figure 12*, HTTP currents were indeed decreased by ~30% immediately after removing extracellular K to deactivate Na/K pumps. Furthermore, the HTTP currents routinely

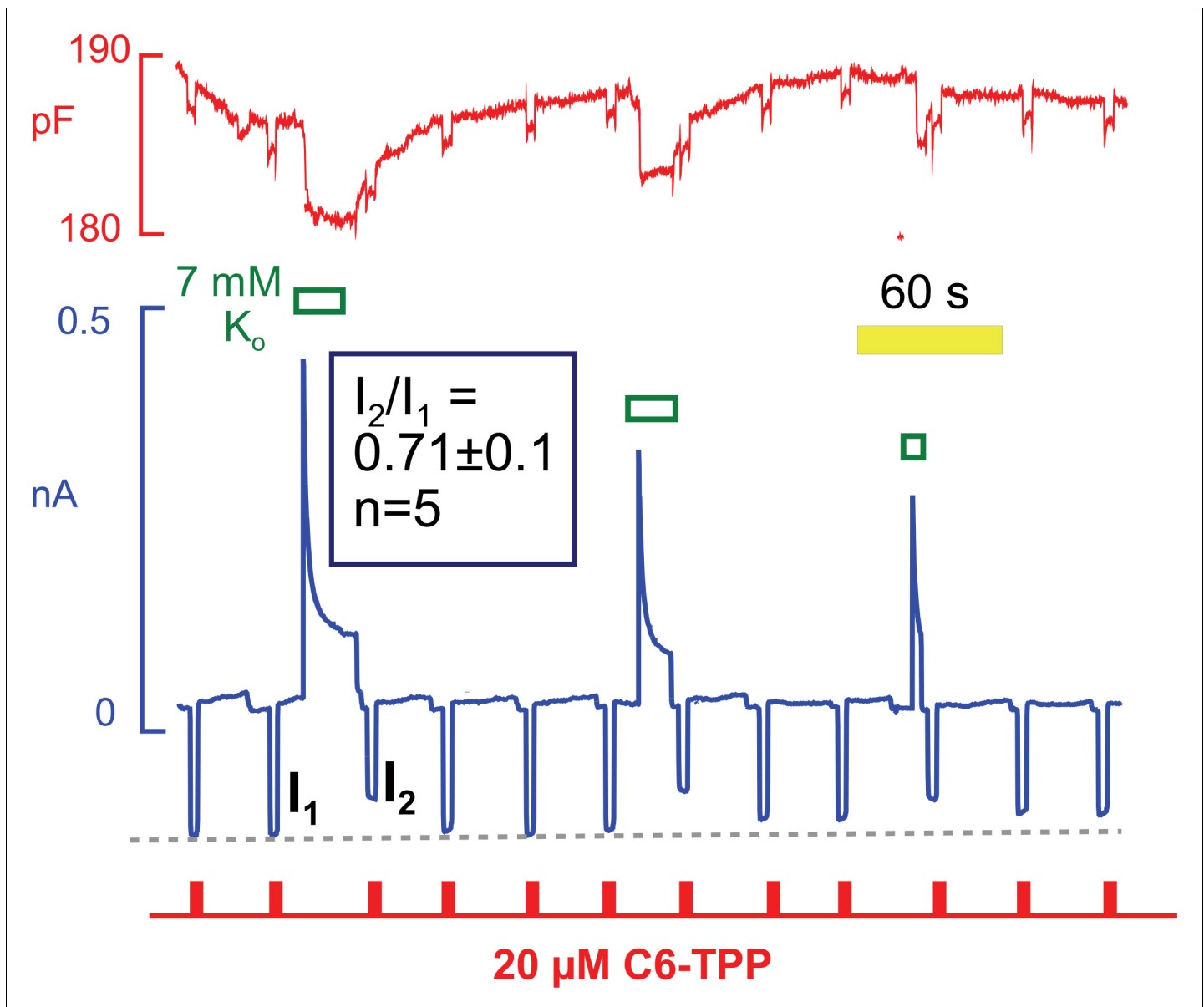

**Figure 12.** C6-TPP currents decrease in response to the apparent inactivation of Na/K pumps. C6-TPP currents were repeatedly activated for 2 s, and Na/K pump currents were activated multiple times between a series of C6-TPP currents. The C6-TPP currents were decreased on average by 29% subsequent to pump activity and current decay, and the C6-TPP current recovered with a time course very similar to the recovery of pump activity and membrane capacitance.

recovered with a time course that was similar to the recovery of pump current from inactivation. Note that the recovery of capacitance also followed this time course. In this context, we point out that Na/K pumps are present in the mouse sarcolemma at a high density. Pump density must be >2000 per $\mu m^2$ to account for maximal currents of 4 pA/pF with maximal transport rates of ~300 $s^{-1}$ (*Gadsby and Nakao, 1989*). Assuming that pumps have diameters of 3 to 4 nm, they will constitute 3 to 5% of the membrane surface area. It is therefore plausible that the conformational state of pumps could influence bulk physical properties of sarcolemma, in particular if pumps are interconnected by a protein network, such as cytoskeleton.

The finding that hydrophobic cation currents are inhibited by Na/K pump activity, and recover as pump currents recover, is equivalent to results for Na/Ca exchange currents that support the occurrence of subsarcolemmal Na depletion (*Su et al., 1998*). As verified in *Figure 13A*, and as expected for depletion of cytoplasmic Na, outward Na/Ca exchange current (i.e. Na efflux driven by Ca influx) is suppressed immediately after a period of high Na/K pump activity, and exchange current recovers with a time course that is similar to the recovery of pump currents . In 10 similar experiments, cytoplasmic free Ca was strongly buffered with 25 mM EGTA to 0.2 $\mu M$, and reverse Na/Ca exchange currents were activated repeatedly by applying extracellular Ca (5 mM) for 2 to 3 s before and after activating Na/K pump currents. The extracellular solutions contained 25 mM Na to block pump current that might be activated by contaminating K. Exchange currents were decreased immediately after the pump current transient by a fractional amount that was about one-half of the fractional decay of pump current. Exchange currents then recovered with a time course that was consistently very similar to the time course with which Na/K pump current availability recovered. Based on pump currents, the apparent volume in which Na depletion would be taking place amounted to 2.2 ± 0.5 pL (n=9).

While these results are highly consistent with Na depletion affecting Na/ Ca exchange currents, the previous results suggest that Na/K pump inactivation could affect Na/Ca exchange function by other mechanisms. Therefore, we present one more test for the existence of a restricted cytoplasmic Na space in myocytes in *Figure 13B*. In these experiments we tested whether Na influx by *forward* Na/Ca exchange can enhance Na/K pump currents, as expected for accumulation of Na in a space shared by exchangers and pumps. As described in *Figure 13B*, large inward Na/Ca exchange currents were generated using cytoplasmic solutions in which the free cytoplasmic Ca was heavily buffered to 0.5 $\mu M$ (50 mM EGTA with 35 mM Ca). In this way, continuous inward exchange currents are generated and can be switched on and off by applying and removing nickel (Ni, 4 mM) in the extracellular solution. We note that Ni blocked isolated exchange currents within our solution switch times, and we note also that 5 mM Ni blocked equally rapidly and completely reversibly significant fractions of the Na/K pump current in mouse myocytes. This latter result is notably different from reports for myocytes from other species (*Fujioka et al., 1998*). That Ni is blocking primarily inward Na/Ca exchange current in the protocols employed here is supported by the fact that Ni caused no outward current shifts under routine experimental conditions when pump activity was not activated (10 observations).

As described in *Figure 13B*, pump currents were activated with extracellular K after prolonged (30 s) inhibition of inward exchange currents *and* during the continued activation of inward exchange currents. The peak pump currents were in fact significantly smaller in the presence of Na influx via Na/Ca exchange than after prolonged blockade of inward exchange current. Composite results for 9 experiments are given below the pipette cartoon in *Figure 13B*. To highlight the contradiction that emerges, simulated results for a restricted Na space model are shown in *Figure 13C*. In brief, the simulation was set up conservatively, using a hyperbolic Na dependence of pump current on cytoplasmic Na with a $K_d$ of 5 mM (i.e. with a stong tendency to saturate at the 10 mM $Na_i$ concentration employed in experiments). With cytoplasmic space reduced enough to recreate pump current decay, the peak Na/K pump current should have been substantially increased in the presence of inward exchange current that amounts to 30% of the peak pump current and that moves 3 Na per elementary charges inwardly per cycle.

## Discussion

We have described pronounced Na/K pump current changes in intact cardiac myocytes that may reflect depletion of cytoplasmic Na in a restricted subsarcolemmal space *or* the function of an

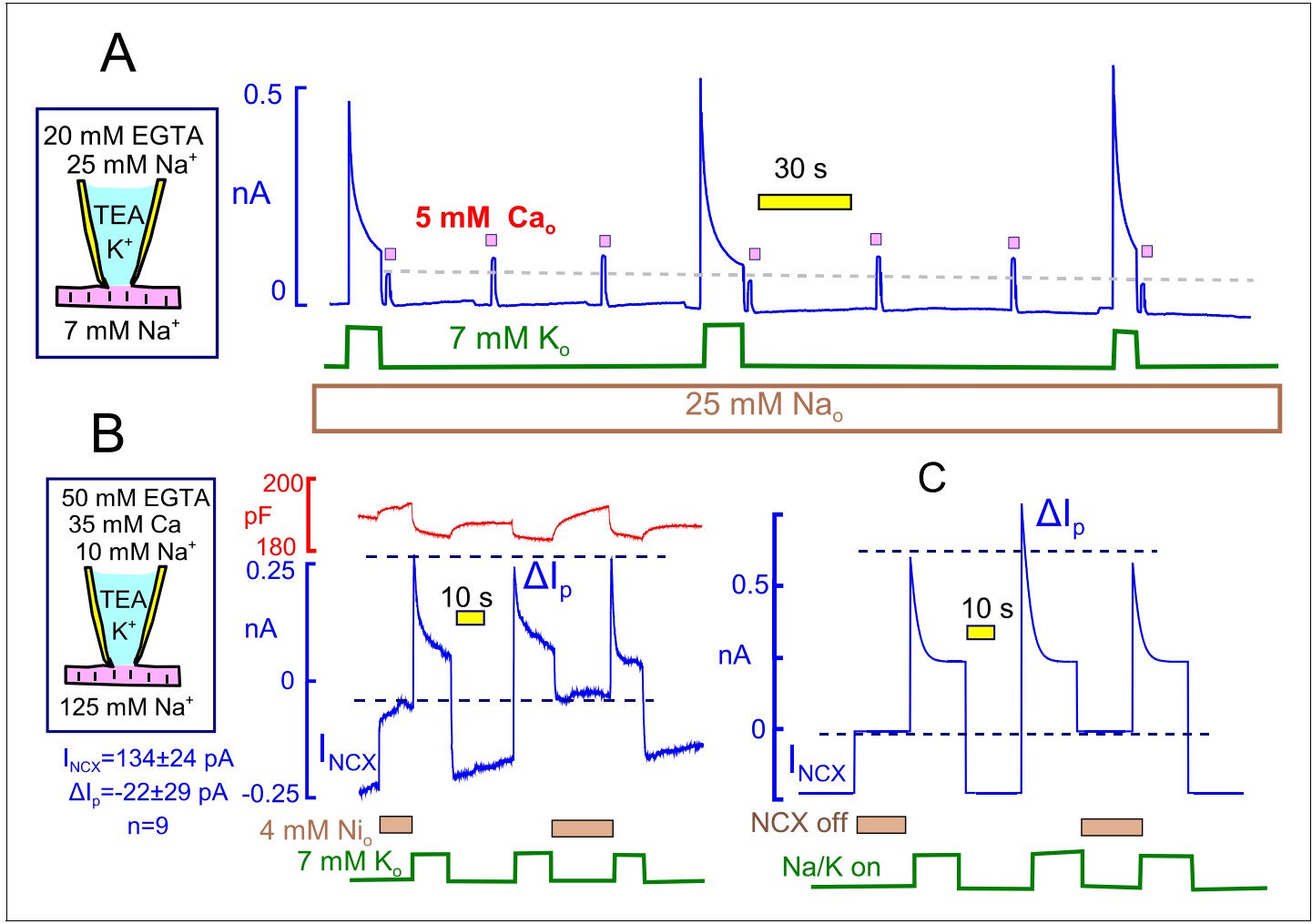

**Figure 13.** Evidence for and against Na/K pump - Na/Ca exchange cross-talk via submembrane Na changes. (**A**) Standard cytoplasmic solution was modified to effectively buffer cytoplasmic free Ca to ~0.2 µM (25 mM EGTA + 6 mM Ca). Reverse Na/Ca exchange currents were activated repeatedly for 2 s before and after Na/K pump currents were activated for 15 s. Exchange currents, like C6-TPP currents in *Figure 12*, are suppressed immediately after pump activity and recover over 20 s. (**B**) K-free cytoplasmic solution was buffered to contain 0.5 µM free Ca, thereby supporting a large, continuous inward exchange current (i.e. Na influx and Ca extrusion). 50 mM $EGTA_i$+ 35 mM $Ca_i$, 10 mM $Na_i$, and 125 mM $Na_o$. Inward exchange current was initially blocked, as indicated, by applying 4 mM Ni on the extracellular side. Pump current was then activated by extracellular K and exchanger block was released by removing Ni at the same time. Subsequently, Na/K pump current was activated once without and once with Na/Ca exchange block. The presence or absence of inward exchange current, before activation of pump current, does not affect the magnitude of peak pump currents. Composite results for 9 experiments are tabulated below the cartoon. (**C**) Predictions of a simple $Na^+$ depletion model. Pump currents were simulated with a hyperbolic dependence on cytoplasmic $Na^+$ ($K_{50}$, 5 mM), and the cytoplasmic space was limited to a value of 3 pL to promote submembrane Na changes. When Na depletion during pump activity limits pump currents and suppresses reverse exchange currents, as in Panel **A**, forward exchange currents *should* very significantly enhance pump currents under the conditions of Panel **B**, and that is not the case.

autoregulatory, inactivation mechanism. Our results do not definitively negate the local Na depletion hypothesis. However, the great majority of results favor an inactivation mechanism that occurs when transport conditions promote occupancy of E1 states with at least one unoccupied Na binding site. Most importantly, transient Ca elevations strongly attenuate current decay and *appear* to increase the affinity for cytoplasmic Na. The stimulatory effects of Ca elevations, as well as the apparent inactivation itself, occur *hand-in-hand* with changes of physical-chemical properties of the sarcolemma, sensed by nonselective membrane probes. We discuss these results first in relation to the physiological function of cardiac Na/K pumps and subsequently in relation to mechanistic questions that arise for both the depletion and inactivation hypotheses.

## Physiological regulation of cardiac Na/K pumps by Ca-dependent processes

The fundamental result of this study is that Ca influx, usually associated with cycles of internal Ca release, can strongly up-regulate cardiac Na/K pump activity by mechanisms that do not seem to involve protein kinases or redox signaling. The functional consequence will presumably be a down-regulation of basal cardiac contractility, thereby completing a negative-feedback loop in the regulation of cardiac excitation-contraction coupling. Na/K pumps are very well poised to fulfill this function: Na/K pump activity in the presence of cytoplasmic K has a steep dependence on cytoplasmic Na (Hill coefficients > 2; *Figure 2*). Therefore, Na/K pumps minimize changes of cytoplasmic Na whenever Na influx increases or decreases. By minimizing the influence of Na influx, and therefore of Na-conducting ion channels, Na/K pumps emerge as singularly powerful regulators of basal cardiac contractility (*Hilgemann, 2004*).

As noted in the Introduction, previous studies already suggested that cytoplasmic Ca is in some way essential in the signaling cascade by which cAMP-dependent protein kinases enhance pump activity (*Gao et al., 1996*). Our results now raise the possibility that Ca transients regulate cardiac Na/K pumps independent of cAMP, and that cAMP-dependent kinases will to a *greater-or-lesser* extent up-regulate Na/K pump activity secondary to their well- known actions to enhance myoycte Ca transients. In the presence of saturating cytoplasmic Na, pump currents are not increased by Ca elevations (*Figure 6B*). Thus, attenuation of the *putative* inactivation reaction by Ca elevations duplicates the apparent increase of Na affinity ascribed to phospholemman phosphorylation (*Han et al., 2010*). An interesting parallel to our results is the description of Na/K pump activation in skeletal muscle by a train of rapid excitations, reviewed recently (*Clausen, 2013*). Although extracellular K definitely accumulates in skeletal muscle T-tubules during rapid action potential firing, and thereby promotes Na/K pump activity (*DiFranco et al., 2015*), this mechanism cannot explain why ouabain-sensitive rubidium uptake is strongly enhanced by skeletal muscle activity (*Buchanan et al., 2002*). It seems reasonable to suggest therefore that the Ca-dependent activation of Na/K pump activity described here in cardiac myocytes is physiologically important in skeletal muscle as well.

Disappointingly, our work provides no insight into the role of phospholemman phosphorylation per se by PKAs, as we observe no relevant changes in Ca-dependent modulation of pump function. At the same time, our work gives no reason to doubt that phospholemman phosphorylation increases the effective Na affinity of Na/K pump ATPase activity in isolated membranes, as described by multiple authors with multiple methods (*Bibert et al., 2008*; *Manoharan et al., 2015*; *Mishra et al., 2015*). One potentially important regulatory factor that requires future attention is the palmitoylation of phospholemman at two cysteines, a modification that appears to be required for phospholemman to inhibit pump activity (*Tulloch et al., 2011*). In this connection, it is doubtful that phospholemman is heavily palmitoylated under the conditions of our experiments (*Lin et al., 2013*).

We have at this time only limited insight into the pronounced inhibitory effects of extended or repeated cytoplasmic Ca elevations on Na/K pump currents (*Figures 8* and *10*) that we observe very routinely. A Ca-dependent generation of fatty acids and lysolipids would potentially inhibit pump activity by direct actions (*Swarts et al., 1990*). A combined generation of acyl CoAs and activation of PKCs has been shown to promote palmitoylation of membrane-associated proteins, including phospholemman (*Lin et al., 2013*) The opening of mitochondrial permeability transition pores, which also can occur in these protocols (*Hilgemann et al., 2013*; *Lin et al., 2013*), might promote reverse F-ATPase activity and thereby cause depletion of cytoplasmic ATP (*Bernardi et al., 2015*).

## The conundrum of Na/K pump current decay: Inactivation versus cytoplasmic Na depletion

Biophysical analysis of cardiac myocyte ion homeostasis during patch clamp suggested that cytoplasmic Na concentrations would not deviate by more than 5% from Na concentrations employed in pipette solutions (*Mogul et al., 1989*). Nevertheless, multiple groups were able to measure significant changes of cytoplasmic Na in myocytes during activation of Na/K pump currents (*Su et al., 1998*; *Despa and Bers, 2003*), albeit on longer time scales than current decay in the present experiments. Thus, the faster components of Na/K pump current decay have remained open to other interpretations. The fact that Na/K pump activity causes a parallel decline of Na/Ca exchange activity (*Figure 13A*) is a very persuasive argument for Na depletion, although the present work suggests a

plausible alternative. Given that hydrophobic cation currents decrease in parallel with Na/K pump current decay (*Figure 12*), it is likely that significant membrane changes are occurring and the possibility thus arises that Na pump inactivation might regulate Na/Ca exchange function by mechanisms that are independent of Na concentration changes. That Na/K pumps might physically influence Na/Ca exchangers within protein complexes was suggested already by *Su et al. (1998)* in this same context. This might be possible if inactivation involves unusual physical changes of pump structure, as might occur with dimerization, and that might mechanically influence the bilayer as well as interacting protein networks. Alternatively, inactivation might control an enzymatic activity that alters the membrane, similar to the long-standing (*Tian et al., 2006*) but still controversial (*Yosef et al., 2016*) hypothesis that Na/K pumps directly regulate Src kinases.

The measurement of Na/Ca exchange reversal potentials in equivalent experiments with guinea pig myocytes (*Fujioka et al., 1998*) should have distinguished between the major two hypotheses, but the outcomes were ambivalent. Using highly Ca-buffered pipette solutions, activation of Na/K pumps by extracellular K caused a time-dependent decrease of Na/Ca exchange currents with no change of reversal potential, as expected if Na/K pumps physically regulate Na/Ca exchangers. However, Na/Ca exchange reversal potentials changed markedly during recovery from pump activity, as if subsarcolemmal Na had been depleted during pump activity and recovered after termination of pump activity. Clearly, this issue requires further work. The measurement of Na channel reversal potentials might be more decisive than Na/Ca exchange reversals because the stoichiometry of NCX1 Na/Ca exchange operation is not completely fixed and may be subject to regulatory changes (*Kang and Hilgemann, 2004*).

While not definitive without energetic proof (i.e. reversal potential measurements), all other results of this study favor the idea that Na/K pump current decay reflects primarily an inactivation mechanism: (1) Capacitance changes that arise with good certainly with Na/K pumps reverse slowly after pump activity is terminated (*Figures 2B*,*3B*,*5A*,*6A,* and *9*). Thus, Na/K pumps probably do not immediately return to an E2 configuration when pump activity is terminated. (2) Na/K pump current decay occurs to the same fractional extent at 23 and 37°C, although pump currents are about four-times smaller at the lower temperature (*Figure 3*). (3) The inclusion of cytoplasmic K in pipette solutions decreases the apparent cytoplasmic Na affinity and thereby decreases pump current densities when Na is submaximal (*Figure 2C and D*). Nevertheless, the fractional decay of pump currents is increased (*Figure 3*). (4) Current decay is very strong when a high Li concentration (40 mM) is used as a Na surrogate and pump currents are three-fold smaller than in our standard conditions (*Figure 3*). (5) We have verified a key prediction for the inactivation model, suggested in *Figure 1*. Application of extracellular Na under conditions that promote accumulation of cytoplasmic ADP suppresses the transient phase of pump currents, such that pump currents activate to approximately their quasi steady state (15 s) magnitude without a peak followed by current decay (*Figure 5*). (6) Myocytes can support outward Na/Ca exchange currents that are many times larger than pump currents (*Figure 7*). Within the context of a Na depletion model, one must assume that Na/Ca exchangers have access to a much larger restricted space than Na/K pumps in those myocytes, or alternatively that the restricted space and/or Na diffusion rates are strongly enhanced within a very short time by Ca influx. (7) A large, continuous Na influx via forward Na/Ca exchange fails to enhance pump currents, although the operation of pumps can depress reverse exchange currents (*Figure 13B*). (8) The fractional decay of pump currents often increases as currents decline over time, a behavior that is opposite to the pattern expected for an ion depletion mechanism (e.g. *Figure 11*). Finally, as noted in Results, the rate of pump current decay does not decrease as cytoplasmic Na is increased into the saturation range for pump activity (*Verdonck et al., 2004*).

Apart from Na concentration changes, the circumstances in which ATP depletion, coupled with ADP and Pi accumulation, might support current decay require future attention. During a maintained average pump current of 300 pA, about 2 mmole ATP per liter cytoplasm will be cleaved every 10 s in a 20 pL cell. In the physiological setting, these nucleotide changes must be very rapidly countered by ATP regeneration via glycolysis and/or oxidative phosphorylation. On the one hand, extensive local depletion of ATP might be an immediate cause of pump current decay during metabolic stress. On the other hand, our work suggests that ADP accumulation during metabolic stress will favor occupancy of states that can inactivate. Independent of Na/K pump function, it appears that significant membrane changes are occurring rapidly in the presence of metabolic stress (*Figure 11*).

## How does cytoplasmic Ca elevation enhance Na/K pump activity?

Transient elevations of cytoplasmic Ca have two effects when cytoplasmic Na is non-saturating. Ca elevation decreases the Na/K pump current decay *and* increases peak pump currents (*Figures 6–10* and *Table 1*). In both a 'Na depletion' model *and* an 'inactivation' model, a genuine increase of Na/K pump Na affinity will of course cause an increase of peak pump currents when extracellular K is applied. Qualitatively, a genuine increase of Na affinity can also explain changes of pump current decay that occur. In a depletion model, cytoplasmic Na must decrease more extensively to cause a fall of pump current when the Na affinity is increased. In an inactivation model, a higher Na affinity attenuates inactivation because the number of E1 pumps without a bound Na would be decreased. In this context, the existence of a labile, hydrophobic Na antagonist would go quite far to explain many results. Nevertheless, it is not necessary to evoke a genuine change of Na affinity to explain these results. One possible extension of the inactivation model is that pumps inactivate at multiple points in the pump cycle, including E2 states. If transient Ca elevation favors the recovery from *multiple* inactive states, not just those immediately coupled to E1 conformations, peak pump currents will increase without evoking a genuine increase of Na affinity.

In a recent study employing rabbit cardiac myocytes, Na/K pump current decay during application of extracellular K was implicated to involve the glutathionylation of Na/K pump beta subunits (*Garcia et al., 2016*). For unknown reasons, our equivalent results with mouse myocytes are very different, and we stress that current decay in mouse myocytes is much faster than in rabbit myocytes. As described in *Table 1*, we have tested quite extensively whether Na/K pump activity is regulated by redox-dependent processes, including glutathionylation. Experimental results for our routine protocols are similar in myocytes from mice lacking Peroxyredoxin6 and in which turnover of oxidized glutathione to glutathione should be decreased (*Zhou et al., 2013*). Experiments employing high concentrations of cytoplasmic glutathione (>5 mM), oxidized glutathione (>5 mM), superoxide dismutase, and reducing agents give no indication that redox-dependent processes are *immediately* involved in the effects of Ca elevations or their reversal (*Table 1*).

Our efforts to identify specific molecular mechanisms by which Ca elevation modulate Na/K pump activity are to date unsuccessful (*Table 1*). These negative outcomes beg the suggestion that important Ca-dependent regulatory systems remain to be elucidated in cardiac myocytes. It may be an important 'clue' that Ca elevation and Na pump inactivation cause significant but opposite physical changes of the bulk surface membrane, as detected with hydrophobic ions. Further, it appears important that the effects of Ca elevation in intact myocytes have no clear parallels in biochemical pump studies. And finally, we stress that we have never observed comparable effects of cytoplasmic Ca on Na/K pump currents in giant excised membrane patches in which pump currents appear to be highly activated in comparison to pump currents in intact cells (*Lu et al., 1995*). One reason could be that giant excised patches are highly stretched membranes which probably lack actin membrane cytoskeleton. On the other hand, pump currents in excised patches routinely show an initial 'run-up' that is consistent with the irreversible loss of an endogenous pump inhibitor (*Hilgemann, 1997*). With certainty, critical factors are disrupted or lost when membranes are isolated.

## Summary and future direction

Our data reveal three parallel changes of Na/K pump and Na/Ca exchange activities that can be tracked by a nonselective membrane probe, C6-TPP: (1) In response to a transient Ca elevation, both pump currents and exchange currents usually become facilitated (*Figure 8B*). (2) In response to repeated transient Ca elevations, both currents become suppressed and/or are ablated (*Figure 8C* and *10A*). (3) Na/K pump current decay during application of extracellular K is associated with a substantial decline of Na/Ca exchange current (*Figure 13A*). The fact that membrane probes sense all three events supports speculation that the three events are related.

Increases of membrane capacitance that occur very reliably after Ca elevations (*Figures 6,9,10*) indicate that membrane fusion events are occurring and/or that the membrane is becoming slightly thinner. Since maximal pump currents are not increased by transient Ca elevation (*Figure 6B*), insertion of pumps is not a viable explanation for the activation of pump currents. However, Ca-dependent fusion events could bring into the cell surface proteins that subsequently mediate pump activation. The parallel increase of hydrophobic ion signals (*Figure 1*) indicates that physical properties of the membrane are indeed changing. Specifically, the membrane is becoming

more disordered. Ca-dependent enzymes that might mediate such changes include PLA2s (*Brown et al., 2003*), diacylglycerol kinases (*Sakane and Kanoh, 1997*), and acyl transferases that generate acyl-phosphatidylethanolamine (*Ogura et al., 2016*).

An alternative is that cardiac Na transporters in intact myocytes physically regulate one another through a protein network that involves the membrane bilayer itself. Cardiac (NCX1) Na/Ca exchangers have a large cytoplasmic regulatory domain with multiple Ca binding sites that effectively responds to changes of the time integral of cytoplasmic free Ca (Matsuoka, 1993). Both conformational changes per se of this domain (*John et al., 2011*) and its activating influence on Na/Ca exchange function (*Hilgemann et al., 1992a*) *can* persist for longer than one minute after cytoplasmic Ca declines. Thus, the cytoplasmic NCX1 domain is one potential Ca sensor for the activation of Na/K pumps by Ca elevation. Conformational information would be transferred to Na/K pumps via cytoskeleton, the bilayer, and/or an enzymatic activity. The existence of physical interactions between neighboring transporters, independent of Na concentration changes, is an attractive explanation for findings that specific Na/K pump isoforms regulate cardiac excitation-contraction coupling in a specific manner (*Rindler et al., 2013*). As concerns the Na depletion hypothesis, the physical basis of a restricted subsarcolemmal Na space with multi-second Na exchange remains to be established. We stress that our results do not at this time rigorously exclude its existence. New model systems, which can mimic the functions of Na/K pumps in intact myocytes, are now essential to resolve many questions opened by this study.

In conclusion, we have initiated a new effort to identify physiological mechanisms that regulate Na/K pump activity in cardiac myocytes. Surprisingly, Ca elevations modulate cardiac Na/K pump activities more powerfully than any conventional signaling pathway in our experience. Transient Ca elevations attenuate Na/K pump current decay by mechanisms that remain to be elucidated. Common Ca-dependent protein kinases and phosphatases do not appear to be involved, and the underlying mechanisms result in physical changes of the membrane that can be detected by multiple nonspecific membrane probes. The Ca-dependent mechanisms that modulate Na/K pumps may also modulate the function of Na/Ca exchangers, and the involvement of membrane per se suggests that other membrane-coupled processes might also be affected.

## Materials and methods

### Electrical methods and myocytes

Patch clamp (*Yaradanakul et al., 2008*) and myocyte preparation were as described (*Lariccia et al., 2011*). The UT Southwestern Medical Center Animal Care and Use Committee approved all animal studies. Highly polished pipette tips with diameters of >4 μm were employed, and access resistances during recordings ranged from 1.2 to 4 MΩ. All experiments presented were performed at 0 mV.

### Fluo4 Ca measurements

Freshly isolated cardiac myocytes were incubated with 2.6 μM Fluo-4 AM (Thermo Fisher Scientific) at 23°C for 60 min. Epifluorescence imaging was performed with a Nikon Eclipse TE2000-S microscope equipped with a Photometrics Cool Snap ES2 camera and a 40 × WI objective. A 470/40 nm excitation filter was employed with a 495 dichroic and 500 LP emission filter set. Analysis was performed using Nikon NIS Elements AR 4.50.00 64-bit after background subtraction.

### Solutions

Standard Solutions employed minimize all currents other than Na/K pump currents and Na/Ca exchange currents. The standard extracellular solution contained in mM: 110 N-methyl-D-glucamine (NMG), 4 $MgCl_2$ ± 2 $CaCl_2$, 0.5 EGTA, 20 TEA-OH, 7 NaCl or KCl, and 10 HEPES, set to pH 7.0 with aspartate. The standard cytoplasmic solution contained in mM: 90 KOH, 20 TEA-OH, 25 Na-OH, 15 HEPES, 0.5 $MgCl_2$, 0.5 EGTA, 0.25 $CaCl_2$, 1 $K_2PO_4$, set to pH 7.4 with aspartate. Unless stated otherwise, 8 mM MgATP, 2 mM TrisATP, and 0.2 mM GTP were employed in cytoplasmic solutions, generating a free $Mg^{2+}$ of 0.5 mM. When different monovalent cations were employed in the cytoplasmic solution, they were added as hydroxides, and NMG was used as the cation substitute in experiments varying monovalent cations. Solutions employing 20 and 50 mM EGTA were prepared

by first dissolving EGTA with NMG to give a neutral pH, followed by boiling with the required amounts of $CaCO_3$. The final osmolarity of all solutions was 290 mosmol/L. In experiments examining the cytoplasmic Na dependence of pump currents with K, K was reduced from 90 to 70 mM in the K-containing solutions employed, and NMG was substituted for Na. NMG was substituted for cytoplasmic K in experiments without K.

## Reagents and chemicals

All salts were from Sigma-Aldrich and were the highest available grade. Reagents employed in experiments described in *Table 1* were from standard chemical suppliers.

## Mice

PLM-deficient mice were bred from knockout mice provided by Amy L. Tucker (U.Virginia, Charlottesville) (*Jia et al., 2004*). PLD1/2-deficient mice were bred from knockout mice provided by Michael Frohman (Stony Brook U., Stony Brook, NY) (*Ali et al., 2013*). DHHC5-deficient cardiac myocytes were isolated from littermates of heterozygous crosses at F2 or littermates of F2 x F2 crosses of homozygous WT or DHHC5-Gene-Trapped mice (*Li et al., 2012*). PRD6-deficient mice were bred from knockout mice provided by Aron B. Fisher (U. Pennsylvania, Philadelphia) (*Nagy et al., 2006*)

## Simple simulations employed in *Figure 1E*

The simulated results were generated by assuming that peak pump currents (Ipeak) are proportional to the simultaneous binding of Na to three sites, two of which have the same affinity:

$$I_{peak} = I_{max} \cdot (Na_i/(Na_i + Kn_1))^2 * Na_i/(Na_i + Kn_2), \tag{1}$$

where $Kn_1$ was 0.5 mM and $Kn_2$ was 4.0 mM. In the case of a restricted space (dotted curves in *Figure 1E and F*, the steady state current ($I_{ss}$) was calculated by assuming that Na next to the membrane could decrease toward a steady state value ($Na_{ss}$) from a constant bulk Na concentration ($Na_B$) in dependence on a single diffusion barrier ($K_{diff}$). Accordingly, the net Na flux through the restriction is equal to the Na flux out of the cell mediated by pumps:

$$I_{ss} = I_{max} \cdot (Na_{ss}/(Na_{ss} + kn_1))^2 \cdot Na_{ss}/(Na_{ss} + Kn_2) = (Na_B - Na_{ss}) \cdot K_{diff}, \tag{2}$$

where $Na_B$ is the bulk cytoplasmic Na concentration, $Na_{ss}$ is the concentration at the membrane, and diffusion into the restricted space is determined by $K_{diff}$(0.06 pA•pF$^{-1}$•mM$^{-1}$). Equation #2 was solved for $Na_{ss}$ by a Newton procedure and the steady state current, $I_{ss}$, was calculated accordingly.

For the inactivation model results (solid curves in *Figure 1E and F*), we assumed that pumps inactivate in proportion to the fraction of pumps in which the $Kn_2$ site is not occupied, $F_o = Kn_2/(Kn_2 + Na_i)$. Assuming that recovery from inactivation occurs at 0.3 s$^{-1}$ and is independent of Na binding, and assuming additionally that inactivation takes place at a rate of $F_o \cdot 1s^{-1}$, the steady state currents ($I_{ss}$) was calculated as:

$$I_{ss} = I_{peak} \cdot 0.3/(F_o + 0.3), \tag{3}$$

## Statistics

Unless stated otherwise, error bars represent standard errors. Significance was assessed by Students T-test or, in occasional cases of inappropriate variance differences, by the Mann-Whitney Rank Sum test. Maverick data points were removed from data sets by the criterion that a data point deviated by more than *two standard deviations* from the data set mean.

## Acknowledgements

We thank Mei-Jung Lin (UTSW) for technical assistance, Michael Fine (UTSW) for discussions, and David C Gadsby (Rockefeller) for constructive criticism.

## Additional information

### Funding

| Funder | Grant reference number | Author |
|---|---|---|
| National Institutes of Health | RO1 #1129843 | Donald W Hilgemann |
| Charles and Jane Pak Center of Mineral Metabolism and Clinical Research | Endowed Professor Collaborative Research Support | Donald W Hilgemann |
| American Heart Association | Fellowship #30950013 | Christine Deisl |

The funder (NIH) had no role in study design, data collection, interpretation, or the decision to submit the work for publication.

### Author contributions

F-ML, Conception and design, Acquisition of data, Analysis and interpretation of data; CD, Acquisition of data, Analysis and interpretation of data; DWH, Conception and design, Acquisition of data, Analysis and interpretation of data, Drafting or revising the article, Contributed unpublished essential data or reagents

### Author ORCIDs

Donald W Hilgemann, http://orcid.org/0000-0002-5288-5133

### Ethics

Animal experimentation: All experiments were performed in strict accordance with the recommendations in the Guide for the Care and Use of Laboratory Animals of the National Institutes of Health. All of the animals were handled according to approved institutional animal care and use committee (IACUC) protocols. The protocol was approved by the Committee on the Ethics of Animal Experiments of the University of Texas Southwestern Medical center at Dallas (Protocol Number: 2015-101114). Euthanasia was performed with flurane and every effort was made to minimize suffering.

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
