## [Decision Letter]

Thank you for submitting your article "Profound Regulation of Cardiac Na/K Pump Activity by Calcium Transients Initiated by Reverse Na/Ca Exchange" for consideration by *eLife*. Your article has been reviewed by three peer reviewers, and the evaluation has been overseen by a Reviewing Editor, David Clapham, and Gary Westbrook as the Senior Editor. The following individual involved in review of your submission has agreed to reveal his identity: David Gadsby (Reviewer #1). The reviewers have discussed the reviews with one another and the Reviewing Editor has drafted this decision to help you prepare a revised submission.

Summary: (from the Reviewing editor)

The reviewers and I agree the paper is pretty much acceptable with minor revisions (no new experiments required). We all agreed that there is already a wealth of data and you do not need to add more, but I have included all three reviewers’ full comments so that you can alter the Results or Discussion to clarify what other readers may have questions about.

The major point that all agree on is that you need to clarify what you mean by calcium transients- sometimes it seems you mean addition of external Ca, sometimes CaV opener, sometimes spontaneous contraction. Please be clear in the text and legends and whether a Ca transient was measured or inferred.

Second, please read below the questions about capacitance changes and address as you think best – again no one is suggesting new experiments, just that you clarify your discussion on this point, perhaps using the questions asked by the reviewers as a guide.

See comments from reviewers 2 and 3 about ATP experiments, Ni, and Li. These require no new experiments, but you may want to alter your Discussion as others may have similar ideas or questions.

Please address the grammar and language – e.g. the use of "Therewith"!

*Reviewer #1:*

There have been many previous attempts to characterize and dissect mechanisms of Na/K pump regulation in the heart. Most have examined a single pathway or pathway component under one or another condition. Over the years this has resulted in a confusing patchwork of loosely connected, and occasionally conflicting, findings that have offered little consistent insight. The present work raises the bar for future studies aimed at this topic by demonstrating the breadth and depth of high-resolution measurements required to draw reliable conclusions.

Here, in the course of examining the causes of Na/K pump current decline during brief periods of exposure of cardiac myocytes to pump-activating extracellular K, Hilgemann and colleagues have uncovered evidence for an inactivation process that appears to regulate the activity of Na/K pumps under roughly physiological conditions. At least as important, they provide evidence for possible modulation of that inactivation by manipulations that raise cytoplasmic Ca concentration. Shockingly, the magnitude of the modulation identifies it as probably the largest physiological regulator of cardiac Na/K pump function found to date. Moreover, the authors' exhaustive search for the underlying Ca-dependent modulatory pathway seems to have ruled out all the usual suspects, forcing them to consider effects mediated by changes in physico-chemical properties of the cell membrane itself.

The authors present a large amount of carefully controlled experimental data, obtained in intact cardiac myocytes with adequately controlled intracellular and extracellular environment and rapid external solution exchange. The data include measurements of cell membrane current and capacitance, or cytoplasmic Ca concentration. Not surprisingly, among the wealth of experimental information are findings that prompt further questions. Obvious ones are what are the origin and mechanism of the extracellular K-induced decrease in the capacitance signal, which is shown in Figure 1 to be ouabain sensitive, and which recovers relatively slowly (10s of sec) after K withdrawal with a time course that seems to mimic both recovery of Na/K pumps from inactivation and recovery of hydrophobic cation current amplitude from K-induced depression (Figure 12)? Could the capacitance signal in the absence of K include a component of charge movement within the Na-bound phosphorylated Na/K pumps that is then lost when K binds and drives the Na/K exchange evident as net outward current? (Those pump charge movements are known to be diminished at low external Na concentration, but might be evident in high-resolution measurements). Could the slow recovery of that capacitance signal reflect unbinding of the last K ion, given the pump's expected μM K affinity in the near absence of external Na, the millimolar concentration of K applied, and the possibility of diffusion-limited spaces containing high-affinity K binding sites (T-system, between sarcolemma and coverslip)? A pump origin for that capacitance component is supported by its abolition by ouabain (Figure 1) and by its doubling when pump cycling current was similarly doubled (Figure 7). If so, the latter parallel changes in the capacitive and stationary pump current signals could imply a temporary increase in the number of functioning pumps following episodes of Ca entry. The authors acknowledge that increased membrane capacitance after Ca entry could reflect membrane fusion events, but argue that the implied overall ~2% increase in membrane area is hard to reconcile with doubling of pump currents. But if the reversibly inserted membrane were specialized – e.g. T-tubular or caveolar – perhaps this might be possible? A little further discussion of ramifications of the clearly documented capacitance signal changes would help the interested reader.

Given the complexity, variety, and sheer abundance of the data, the authors tread carefully through what could otherwise be a minefield in trying to describe them succinctly and clearly, with one exception. The very prominent shorthand phrase "calcium transients" is not clear and is not defined anywhere. Calcium transients are mentioned in the Title, Abstract, Introduction and first sentence of the Results, and the "stimulatory effects of Ca transients" are introduced in the last paragraph of the Introduction and in Figure 7, whose title attributes the stimulation to "a train of excitation-contraction coupling cycles". The text reasonably clarifies that entry of extracellular Ca via reverse Na/Ca exchange triggers a series of contractions by Ca-induced Ca release, which stops on Ca withdrawal; that text does not mention Ca transients. Nor does the next paragraph, though its title announces "Repeated Ca transients…" and it describes effects of "two consecutive Ca influx protocols" and, later, of "a second Ca influx episode". Those latter usages suggest that a Ca transient means an episode of Ca influx, whereas the previous paragraph might be read as implying that a Ca transient is a single Ca-induced Ca release event, many of which occur during a Ca influx episode. The confusion is not helped by phrases like "the first pump current transient after the Ca transient" (subsection “Stimulatory effects of Ca transients do not appear to involve classical signaling pathways”), or "effects of extended Ca transients" (subsection “Physiological regulation of cardiac Na/K pumps by Ca transients”). If the phrase "Ca transient" is retained it will be essential to define it clearly early on (e.g., transient increase of cytoplasmic Ca concentration) and then to use it consistently.

*Reviewer #2:*

This work describes experiments addressing inactivation of Na/K pump currents and regulation of pump activity by Ca transients in myocytes. The experiments are rigorously done, and suggest that Ca transients associated with reverse Na/Ca exchange profoundly regulate pump activity. The results do not offer a clear mechanistic explanation but nevertheless are significant because they extend established ideas about regulation of Ca(i) by Na/K pump activity and show that reciprocal regulation of exchange and pump activity occurs, and further that this regulation involves mechanisms independent of previously examined candidates like kinase activity, phospholemman, and redox changes. The authors also rigorously analyze and highlight discrepancies between data and predictions of models of inactivation due solely to ion depletion. Further, the authors use hydrophobic ions to show that changes in membrane properties accompany regulatory changes of pump activity. The strength of the work is its rigor and demonstration of a potentially quite important phenomenon in cardiac myocyte physiology, as well as in presenting some potential challenges to current ideas. Its weakness stems from the fact that the mechanisms underlying key observations remain largely obscure. Although the work may be criticized as descriptive, the phenomena are novel and will certainly stimulate further work.

1) The authors establish correlations between pump activity and capacitance, e.g. low Na(i) is required for both a capacitance decrease and for pump inactivation in response to K superfusion. The authors suggest that capacitance changes likely reflect physical changes in addition to membrane trafficking, which is an interesting idea based in part on niflumic acid and C6TPP effects on capacitance, but this mechanism and connection is unfortunately somewhat obscure. Still, an important idea raised by the work is the potential physiological significance of modulation of pump activity by Ca(i), and the existence of a potential feedback loop with linkage to capacitance/physical membrane changes. A single exchanger-mediated Ca transient induces capacitance decrease, repetitive Ca transients induce a large increase, and a single large Ca transient (using FPL64176) induces a decrease. This apparent frequency effect is not discussed; is there any potential mechanistic explanation for apparent dependence on frequency of transients versus quantity of Ca(i)? In addition, the niflumic acid-induced capacitance change after Ca transients increased, but was δ C/C changed? This is important with respect to the postulated relationship between Ca regulation and membrane changes.

2) The authors provide several strong lines of evidence that pump inactivation involves mechanism(s) in addition to Na depletion, and data support a model of selective inactivation from E1 Na-unbound states. A potentially more complex role of ATP is also suggested by some experiments, and this could be further discussed or explored. A prediction of the authors' inactivation model is borne out by experiments designed to deplete ATP and suppress the transient overshoot phase but it would seem to be informative to further test whether increasing glucose or including ATP in the pipette blocked or reduced the pump current inactivation, increased the steady-state current, or changed recovery kinetics. It would also seem worthwhile to test whether ATP affected inhibition of pump currents induced by multiple Ca transients.

3) An important argument concerning the selectivity of the pump-exchanger interaction is based on the experiment shown in Figure 13. This involves simultaneous washout of Ni and application of K, but the unblocking kinetics of the exchanger current following Ni removal are not shown (presumably this is fast, but should be shown or noted). The authors should show that are no pre steady-state Ni washout currents that may be additive or occlusive with the pump current. This would strengthen the conclusion that presence or absence of inward exchanger current does not affect the amplitude of pump current as predicted by a depletion model (instead, inhibition of pump currents is seen in the face of enhanced exchange current). The authors state there is no effect of Ni under "routine conditions", is there also no direct effect of Ni on K-induced pump currents?

*Reviewer #3:*

This well written manuscript presents convincing data for a novel and important form of Na/K pump regulation by Ca transients in the heart. Post Ca transient potentiation is clear, important and the data let no doubt about it. Furthermore, kinase and redox regulation seems to produce no effect on pump function or in its regulation by Ca. The effects of membrane modifiers are also important. All adds important data to controversies regarding Na/K pump regulation.

I don't think the evidence for pump inactivation in E1 without bound Na is conclusive enough, but such an inactivation is an attractive hypothesis that would have profound implications.

Given the insensitivity of the Ca effects to so many compounds, irrefutable proof of E1-inactivation should come from inside-out giant patches. This would allow perfect intracellular control eliminating assumptions of depleted volumes. Under the external conditions used here, if the pumps are inhibited in E1 when Na is not present, the rate of activation of Na/K pump current should differ if ATP is applied in the presence of 25 mM Na+i, than if Na+i is applied in the presence of 10 mM ATP. In order to facilitate occupancy of E1, even in the absence of ATP, the experiments could be performed in continuous presence of 1 mM ADP. Also, ideally, the effects of Ca should also be observed in inside out patches.

Small currents: The experiment with 40 mM Li+i may be misleading because the K0.5 for Li+i is above 40 mM (Hemsworth et al., 1997) and thus is equivalent to low Na+i. The currents at the end of the K application in the experiments where pump current was reduced by lowering the temperature are tiny (see also error bars). How well can this current be measured? Why weren't the currents reduced by reducing [K]o? To completely eliminate possible K-current contaminants, relevant when pump currents are tiny, it would have been better to use Cs+o instead of K^+^o, both for maximal and submaximal current activation (currents in K^+^o and Cs+o are identical in the absence of Na+o).

Regarding the table, which shows an outstanding amount of experiments convincingly demonstrating lack of kinase or redox modulation of Ca potentiation, a couple of observations require further discussion. First, the PLM -/- mice show that the currents in the absence of Ca transients, with 25 mM Na+i Fdecay is identical than in wild type mice with 40 mM Na+i both before and after Ca, although in 40 mM Ca there seems to be some reduction in Ipeak, while the PLM-/- mice have a 25% increase. Second, the membrane modifiers show another important observation. F decay was significantly reduced by methylene blue and Triacsin C, although the peak was reduced by 50%. Can these distinct effects of membrane modifiers and PLM -/-point to separate Ca-regulation mechanisms related to [Na]i?

The experiment in Figure 13 has rundown and a change in baseline. Decay in the first and third K-applications differ. Explanation?

---

## [Author Response]

We thank the editor and reviewers for thoughtful and constructive criticism. The manuscript has been carefully revised in accordance with the reviews. We have additionally strived to improve the article beyond the suggestions of the reviewers, especially to make it more accessible. In the same time, we have made experimental progress in several areas, and this is now reflected in the revised article without increasing the number of figures. I summarize first changes that affect the figures and the order of presentation. Then, I elaborate specific changes in response to the review related to a marked manuscript documenting revisions point by point and reviewer by reviewer.

Figure changes to improve the presentation and clarify scientific issues.

1) Presentation of pump model and inactivation hypothesis. The first experimental results presented are current records and capacitance records with content that must be explained in relation to the working hypothesis. Therefore, I present the pump model and our working hypothesis first as Figure 1. This background is then used immediately to help explain experimental data.

2) Simulations. The original article included highly simplified simulations to illustrate that an 'inactivation model' and a 'Na depletion' model can both explain the Na dependencies of peak and quasisteady state pump current. This presentation took substantial space and, as pointed out by Reviewer #1, was not incisive. To simplify and shorten this presentation, while still illustrating the major point, I have moved the simulated curves and data at room T into the first data figure. Now, all Na concentration-current relations are in one figure (Figure 2), and there are no duplicative data figures. The simulation models were changed as suggested by Reviewer #1, no problem or issues, and the description of the simulations has been moved to Material and Methods. Together, these changes streamline the flow of the paper.

3) High Na result. In Figure 2 show a different experimental record with 110 mM cytoplasmic Na. This record better illustrates the usual result that capacitance decreases and increases within solution switch times when pump currents do not decline (inactivate). This then better sets the stage to think about capacitance changes that occur slowing in Figure 2 when pump currents inactivate.

4) Elucidation of the Na- and K-dependence of Na/K pump inactivation. As noted by reviewers, the K affinity of Na/K pumps in the absence of extracellular Na is very high. Nevertheless, we had found that pump currents increased as K was increased in our experiments over the range of 3 to 7 mM, whereby we always exchanged equal concentration of extracellular Na for K. It became clear in the intervening period that this concentration dependence in fact represented the concentration dependence by which extracellular Na blocks pump activity that occurs as the result of contaminating K in typical experimental solutions. This problem has in fact been investigated thoroughly in the past by Rakowski et al., as referenced in the revised article. We now describe this situation carefully with reference to our standard experimental conditions (i.e. with reference to Figure 2), and we include a new panel to our figure that describes effects of extracellular Na on pump currents (Figure 4). In its present format, this figure now demonstrates (1) that extracellular Na at low concentrations rescues pumps from steady state inactivation, which occurs in nominally Na-free solutions as a result of contaminating K. It demonstrates (2) that this effect of extracellular Na occurs with such high affinity that 7 mM Na is adequate to rescue nearly 90% of pumps from inactivation. It demonstrates (3) that this mechanism likely accounts for previous suggestions that Na/K pumps have an extracellular regulatory Na binding site. And it demonstrates (4) that extracellular Na can promote strongly pump inactivation when reverse transport reactions are enabled by high cytoplasmic Na. In summary, this figure now presents in clear fashion some very fundamental progress in understanding pump function.

5) Results for cAMP. In Figure 5, addressing protein kinase modulation of pump currents, we have now included bar graphs for the effects of including a high concentration of cAMP in the pipette solution. The lack of significant effect eliminates a possibility that isoproterenol and forskolin were ineffective because the cAMP system was blocked at the level of adenylate kinase.

6) Ca does not increase maximal currents. In Figure 6, which first shows the stimulation of pump currents by Ca influx via reverse Na/Ca exchange, we have included a panel B to show data for a high (40mM) Na concentration. This data was previously included in the Table 1. In this way, Figure 6 now documents that Ca influx greatly enhances pump currents when Na is non-saturating but does not enhance pump currents when Na is saturating. In this way, it is made more clear that Ca elevations act to increase the apparent affinity for Na. Furthermore, the figure now illustrates better how Ca elevations change pump current decay (inactivation).

7) Exchange currents can be massively larger than pump currents. While removing one figure that was not critical (the simulations), we have added another figure (Figure 7) that provides very important new information relevant to the Na depletion hypothesis. Figure 7 now shows Na/K pump currents and Na/Ca exchange currents in myocytes that overexpress Na/Ca exchangers by more than 5-fold. Maintained (15s) Na/Ca exchange currents are much larger than even the peak Na/K pump currents. In fact, they are nearly as large as can be expected from the access resistance of the patch pipette. Pump currents show strong decay and respond per usual to Ca elevations. It is impossible to suggest that the Na transporters are sharing a restricted cytoplasmic Na space.

8) Documentation of Ca waves. In Figure 8, which presents Ca measurements during the Ca elevation protocol, we now include a fluorescence record for an optical slice through the myocyte. This reveals that the myocyte is generating profuse Ca waves that have no clear representation in the whole-cell fluorescence record. This in turn now substantiates our description of contractile activity: Myocytes begin to contract spontaneously and vigorously after Ca influx is terminated.

Improved referencing. Before addressing the reviews specifically, I point out that we now reference and discuss two thoughtful papers that favored the idea of a restricted Na space. One paper concerned the kinetics of pump current decay (Verdonck et al., 2004) and a second paper addressed the reversal potential of Na/Ca exchange during Na/K pump current decay (Fujioka et al., 1998). I also point out that we have refined figures and increased lettering size so that they are easier to understand and can be reproduced in a somewhat smaller size than in the original version. Finally, I have combed the manuscript for mistakes and found one wrong number in the present Figure 3, the apparent volume of the depletion space using lithium.

*Summary: (from the Reviewing editor)*

*The reviewers and I agree the paper is pretty much acceptable with minor revisions (no new experiments required). We all agreed that there is already a wealth of data and you do not need to add more, but I have included all three reviewers’ full comments so that you can alter the Results or Discussion to clarify what other readers may have questions about.*

*The major point that all agree on is that you need to clarify what you mean by calcium transients- sometimes it seems you mean addition of external Ca, sometimes CaV opener, sometimes spontaneous contraction. Please be clear in the text and legends and whether a Ca transient was measured or inferred.*

The title is changed to comply. After long consideration, we have struck the term "Ca transient" from the paper, except when referring to a physiological E/C coupling cycle. Instead we accurately describe our experimental manipulation as a "transient Ca elevation".

*Second, please read below the questions about capacitance changes and address as you think best – again no one is suggesting new experiments, just that you clarify your discussion on this point, perhaps using the questions asked by the reviewers as a guide.*

I have altered both the description of experiments and our Discussion to consider the capacitance data in more detail and specifically in relation to our hypotheses.

*See comments from reviewers 2 and 3 about ATP experiments, Ni, and Li. These require no new experiments, but you may want to alter your Discussion as others may have similar ideas or questions.*

I have made appropriate changes as described in my responses to each review.

*Please address the grammar and language – e.g. the use of "Therewith"!*

The paper has been revised to eliminate 'therewith'.

*Reviewer #1:*

*There have been many previous attempts to characterize and dissect mechanisms of Na/K pump regulation in the heart. Most have examined a single pathway or pathway component under one or another condition. Over the years this has resulted in a confusing patchwork of loosely connected, and occasionally conflicting, findings that have offered little consistent insight. The present work raises the bar for future studies aimed at this topic by demonstrating the breadth and depth of high-resolution measurements required to draw reliable conclusions.*

*Here, in the course of examining the causes of Na/K pump current decline during brief periods of exposure of cardiac myocytes to pump-activating extracellular K, Hilgemann and colleagues have uncovered evidence for an inactivation process that appears to regulate the activity of Na/K pumps under roughly physiological conditions. At least as important, they provide evidence for possible modulation of that inactivation by manipulations that raise cytoplasmic Ca concentration. Shockingly, the magnitude of the modulation identifies it as probably the largest physiological regulator of cardiac Na/K pump function found to date. Moreover, the authors' exhaustive search for the underlying Ca-dependent modulatory pathway seems to have ruled out all the usual suspects, forcing them to consider effects mediated by changes in physico-chemical properties of the cell membrane itself.*

*The authors present a large amount of carefully controlled experimental data, obtained in intact cardiac myocytes with adequately controlled intracellular and extracellular environment and rapid external solution exchange. The data include measurements of cell membrane current and capacitance, or cytoplasmic Ca concentration. Not surprisingly, among the wealth of experimental information are findings that prompt further questions. Obvious ones are what are the origin and mechanism of the extracellular K-induced decrease in the capacitance signal, which is shown in Figure 1 to be ouabain sensitive, and which recovers relatively slowly (10s of sec) after K withdrawal with a time course that seems to mimic both recovery of Na/K pumps from inactivation and recovery of hydrophobic cation current amplitude from K-induced depression (Figure 12)? Could the capacitance signal in the absence of K include a component of charge movement within the Na-bound phosphorylated Na/K pumps that is then lost when K binds and drives the Na/K exchange evident as net outward current?*

In the fourth paragraph of the subsection “Na/K pump current decay”: We have performed a number of further experiments to understand these Cm signals. We now cite a previous paper in which it was established that low concentrations of Na cause a capacitance signal by binding to high affinity sites in the E2 configuration of the pump, sites that behave competitively with K. We describe in words experiments that show that a large part of this capacitance indeed arises from Na binding to the high affinity K sites. We also have proved in related experiments that residual K is an issue when extracellular Na is removed completely.

*(Those pump charge movements are known to be diminished at low external Na concentration, but might be evident in high-resolution measurements). Could the slow recovery of that capacitance signal reflect unbinding of the last K ion, given the pump's expected μM K affinity in the near absence of external Na, the millimolar concentration of K applied, and the possibility of diffusion-limited spaces containing high-affinity K binding sites (T-system, between sarcolemma and coverslip)?*

In the first paragraph of the subsection “Na/K pump current decay”: Thanks, we agree and have verified that K contamination in our extracellular solutions is very important. We discuss this now within Results and present data showing that K contamination causes pumps to inactivate when all extracellular Na is removed in Figure 4. It now seems fairly certain that the apparent activation of pumps by extracellular Na, described by Garcia et al., may result from extracellular Na blocking pump activity that occurs as a result of μM K contamination. In other words, the contaminating K allows pumps to cycle a bit and become inactivated.

*A pump origin for that capacitance component is supported by its abolition by ouabain (Figure 1) and by its doubling when pump cycling current was similarly doubled (Figure 7). If so, the latter parallel changes in the capacitive and stationary pump current signals could imply a temporary increase in the number of functioning pumps following episodes of Ca entry.*

This is a good idea, but at least on first consideration would seem to be contradicted by the fact that maximal pump activity is unaffected by Ca elevations.

*The authors acknowledge that increased membrane capacitance after Ca entry could reflect membrane fusion events, but argue that the implied overall ~2% increase in membrane area is hard to reconcile with doubling of pump currents. But if the reversibly inserted membrane were specialized – e.g. T-tubular or caveolar – perhaps this might be possible? A little further discussion of ramifications of the clearly documented capacitance signal changes would help the interested reader.*

In the second paragraph of the subsection “Summary and future direction.”: We have extended this discussion somewhat and we point out that membrane fusion events might bring regulatory proteins into the sarcolemma.

*Given the complexity, variety, and sheer abundance of the data, the authors tread carefully through what could otherwise be a minefield in trying to describe them succinctly and clearly, with one exception. The very prominent shorthand phrase "calcium transients" is not clear and is not defined anywhere. Calcium transients are mentioned in the Title, Abstract, Introduction and first sentence of the Results, and the "stimulatory effects of Ca transients" are introduced in the last paragraph of the Introduction and in Figure 7, whose title attributes the stimulation to "a train of excitation-contraction coupling cycles". The text reasonably clarifies that entry of extracellular Ca via reverse Na/Ca exchange triggers a series of contractions by Ca-induced Ca release, which stops on Ca withdrawal; that text does not mention Ca transients. Nor does the next paragraph, though its title announces "Repeated Ca transients…" and it describes effects of "two consecutive Ca influx protocols" and, later, of "a second Ca influx episode". Those latter usages suggest that a Ca transient means an episode of Ca influx, whereas the previous paragraph might be read as implying that a Ca transient is a single Ca-induced Ca release event, many of which occur during a Ca influx episode. The confusion is not helped by phrases like "the first pump current transient after the Ca transient" (subsection “Stimulatory effects of Ca transients do not appear to involve classical signaling pathways”), or "effects of extended Ca transients" (subsection “Physiological regulation of cardiac Na/K pumps by Ca transients”). If the phrase "Ca transient" is retained it will be essential to define it clearly early on (e.g., transient increase of cytoplasmic Ca concentration) and then to use it consistently.*

Title and Figure 8: We have now extensively revised the descriptions of what is happening in myocytes to be accurate. We now describe our protocols as inducing 'transient Ca elevations', not Ca transients, starting with the title of article and continuing throughout. We clarify that myocytes are undergoing Ca waves of spontaneous Ca release by showing a cross sectional optical slice with the Ca signals in Figure 8

*Reviewer #2:*

*This work describes experiments addressing inactivation of Na/K pump currents and regulation of pump activity by Ca transients in myocytes. The experiments are rigorously done, and suggest that Ca transients associated with reverse Na/Ca exchange profoundly regulate pump activity. The results do not offer a clear mechanistic explanation but nevertheless are significant because they extend established ideas about regulation of Ca(i) by Na/K pump activity and show that reciprocal regulation of exchange and pump activity occurs, and further that this regulation involves mechanisms independent of previously examined candidates like kinase activity, phospholemman, and redox changes. The authors also rigorously analyze and highlight discrepancies between data and predictions of models of inactivation due solely to ion depletion. Further, the authors use hydrophobic ions to show that changes in membrane properties accompany regulatory changes of pump activity. The strength of the work is its rigor and demonstration of a potentially quite important phenomenon in cardiac myocyte physiology, as well as in presenting some potential challenges to current ideas. Its weakness stems from the fact that the mechanisms underlying key observations remain largely obscure. Although the work may be criticized as descriptive, the phenomena are novel and will certainly stimulate further work.*

*1) The authors establish correlations between pump activity and capacitance, e.g. low Na(i) is required for both a capacitance decrease and for pump inactivation in response to K superfusion. The authors suggest that capacitance changes likely reflect physical changes in addition to membrane trafficking, which is an interesting idea based in part on niflumic acid and C6TPP effects on capacitance, but this mechanism and connection is unfortunately somewhat obscure. Still, an important idea raised by the work is the potential physiological significance of modulation of pump activity by Ca(i), and the existence of a potential feedback loop with linkage to capacitance/physical membrane changes. A single exchanger-mediated Ca transient induces capacitance decrease, repetitive Ca transients induce a large increase, and a single large Ca transient (using FPL64176) induces a decrease. This apparent frequency effect is not discussed; is there any potential mechanistic explanation for apparent dependence on frequency of transients versus quantity of Ca(i)?*

In the third paragraph of the subsection “Repeated Ca elevations can reactivate or strongly inhibit Na/K pump currents”: I'm afraid that we do not have enough information to consider quantitatively the influence of Ca elevation frequency versus Ca quantity. It is certain that we can enhance pump activity only about 2-times per myocytes. Then, something is exhausted. I have included content more explicitly in the revised version.

*In addition, the niflumic acid-induced capacitance change after Ca transients increased, but was δ C/C changed? This is important with respect to the postulated relationship between Ca regulation and membrane changes.*

In the third paragraph of the subsection “Ca elevations and metabolic stress may act via physical-chemical changes of the sarcolemma”: The exact numbers are available in the text. The deltaC/C changes dramatically, since the NA signal nearly doubles when Cm changes by 4 or 5% .

*2) The authors provide several strong lines of evidence that pump inactivation involves mechanism(s) in addition to Na depletion, and data support a model of selective inactivation from E1 Na-unbound states. A potentially more complex role of ATP is also suggested by some experiments, and this could be further discussed or explored. A prediction of the authors' inactivation model is borne out by experiments designed to deplete ATP and suppress the transient overshoot phase but it would seem to be informative to further test whether increasing glucose or including ATP in the pipette blocked or reduced the pump current inactivation, increased the steady-state current, or changed recovery kinetics. It would also seem worthwhile to test whether ATP affected inhibition of pump currents induced by multiple Ca transients.*

In the last paragraph of the subsection “Na/K pump current decay”: I have included information about effects of ATP concentration in the appropriate place. Unfortunately, the experience is that we have only very poor control of cytoplasmic ATP in these experiments.

*3) An important argument concerning the selectivity of the pump-exchanger interaction is based on the experiment shown in Figure 13. This involves simultaneous washout of Ni and application of K, but the unblocking kinetics of the exchanger current following Ni removal are not shown (presumably this is fast, but should be shown or noted). The authors should show that are no pre steady-state Ni washout currents that may be additive or occlusive with the pump current. This would strengthen the conclusion that presence or absence of inward exchanger current does not affect the amplitude of pump current as predicted by a depletion model (instead, inhibition of pump currents is seen in the face of enhanced exchange current). The authors state there is no effect of Ni under "routine conditions", is there also no direct effect of Ni on K-induced pump currents?*

In the third paragraph of the subsection “Hydrophobic cation currents track changes of Na/K pump and Na/Ca exchange activity”: I now include more details about these experiments. Yes, Ni acts within solution switch times and yes, we have looked at this in detail. As a result of these queries, I have also now included in the description of these experiments the fact that Ni blocks rapidly and reversibly Na/K pump activity in the murine myocytes. This is why we designed these experiments as we did. It is indeed surprising that no one seems to have reported on this issue before. IF this is a species difference, it means there is a real difference in the binding sites of murine pumps (mostly α 1) and those of other species. I appreciate that I should have honestly described this situation in the original article. I was trying to avoid complexity.

*Reviewer #3:*

This well written manuscript presents convincing data for a novel and important form of Na/K pump regulation by Ca transients in the heart. Post Ca transient potentiation is clear, important and the data let no doubt about it. Furthermore, kinase and redox regulation seems to produce no effect on pump function or in its regulation by Ca. The effects of membrane modifiers are also important. All adds important data to controversies regarding Na/K pump regulation.

*I don't think the evidence for pump inactivation in E1 without bound Na is conclusive enough, but such an inactivation is an attractive hypothesis that would have profound implications.*

*Given the insensitivity of the Ca effects to so many compounds, irrefutable proof of E1-inactivation should come from inside-out giant patches. This would allow perfect intracellular control eliminating assumptions of depleted volumes. Under the external conditions used here, if the pumps are inhibited in E1 when Na is not present, the rate of activation of Na/K pump current should differ if ATP is applied in the presence of 25 mM Na+i, than if Na+i is applied in the presence of 10 mM ATP. In order to facilitate occupancy of E1, even in the absence of ATP, the experiments could be performed in continuous presence of 1 mM ADP. Also, ideally, the effects of Ca should also be observed in inside out patches.*

In the last paragraph of the subsection “How does cytoplasmic Ca elevation enhance Na/K pump activity?”: Yes, we agree. We discuss now further the 'giant patch'. Unfortunately, all of these regulatory properties systems appear to be lost in the giant patch. We point out that this can be due to loss of cytoskeleton or some type of regulatory factor that diffuses away. We agree as well that the ADP may critical and that some giant patch experiments may be possible. However, as described in the section noted above, pump currents become very highly activated immediately when ATP is applied the first time. Possibly this effect reflects loss of an endogenous pump inhibitor and we have spent much time pursuing this area. But we have not yet found what is lost! Hopefully soon.

*Small currents: The experiment with 40 mM Li+i may be misleading because the K0.5 for Li+i is above 40 mM (Hemsworth et al., 1997) and thus is equivalent to low Na+i. The currents at the end of the K application in the experiments where pump current was reduced by lowering the temperature are tiny (see also error bars). How well can this current be measured?*

In the ninth paragraph of the subsection “Na/K pump current decay”: I have cited the Hemsworth paper and changed the relevant text to more clearly indicate that Li activated pump currents with very low affinity. This is indeed why we chose to use Li. We can use a high concentration of Li, so it is unlikely to deplete. The current decay cannot be depletion, it must be something else.…. As concerns current magnitude, the murine pump currents at room temperature are just as large as the pump currents of other myocytes routinely studied at 37 C, e.g. guinea pig myocyte pump currents. Certainly, we prefer larger currents, but we are confident that the measurements at RT remain accurate.

*Why weren't the currents reduced by reducing [K]o?*

For these experiments, we needed to change the overall activity of pumps. Changes of extracellular K would be interesting, but they would not probe the same issue: Is current decaydependent on pump activity per se? That said, we now describe very important data relevant to the extracellular K concentration. Specifically, we describe that contaminating K in our solutions is adequate to cause very strong steady state inactivation of pump when extracellular Na is nominally removed. See Figure 4.

*To completely eliminate possible K-current contaminants, relevant when pump currents are tiny, it would have been better to use Cs+o instead of K^+^o, both for maximal and submaximal current activation (currents in K^+^o and Cs+o are identical in the absence of Na+o).*

In the first and twelfth paragraphs of the subsection “Na/K pump current decay”: We describe now the influence of contaminating K at the beginning of results and in connection with Figure 4.

*Regarding the table, which shows an outstanding amount of experiments convincingly demonstrating lack of kinase or redox modulation of Ca potentiation, a couple of observations require further discussion. First, the PLM -/- mice show that the currents in the absence of Ca transients, with 25 mM Na+i Fdecay is identical than in wild type mice with 40 mM Na+i both before and after Ca, although in 40 mM Ca there seems to be some reduction in Ipeak, while the PLM-/- mice have a 25% increase. Second, the membrane modifiers show another important observation. F decay was significantly reduced by methylene blue and Triacsin C, although the peak was reduced by 50%. Can these distinct effects of membrane modifiers and PLM -/-point to separate Ca-regulation mechanisms related to [Na]i?*

The difference between high Na and PLM may be that the exchange current is bigger with high Na than in our control conditions. This larger amount of Ca influx may cause more inhibitory effects. The second point is that the form of the Na current changes. So yes, in principle that may multiple mechanisms. I feel however that discussion of these point would exceed any reasonable limit on the volume of this manuscript. The data are available to the reader to analyze and think about. We are very happy that the reviewers have found the data to be of interest.

*The experiment in Figure 13 has rundown and a change in baseline. Decay in the first and third K-applications differ. Explanation?*

The statistics document that this is a representative experiment. Admittedly, there are some small changes going on in these experiments with free Ca clamped to 0.5 μM.